# Extensible Immunofluorescence (ExIF) accessibly generates high-plexity datasets by integrating standard 4-plex imaging data

Ihuan Gunawan [1,2], Felix V. Kohane [1], Moumitha Dey[1], Kathy Nguyen[1], Ye Zheng[1,3], Daniel P. Neumann [1], Fatemeh Vafaee [4,5,6], Erik Meijering[2,5,6] & John G. Lock [1,3,5,6] ✉

Standard immunofluorescence imaging captures just ~4 molecular markers (4-plex) per cell, limiting dissection of complex biology. Inspired by multimodal omics-based data integration approaches, we propose an Extensible Immunofluorescence (ExIF) framework that transforms carefully designed but easily produced panels of 4-plex immunofluorescence into a unified dataset with theoretically unlimited marker plexity, using generative deep learning-based virtual labelling. ExIF enables integrated analyses of complex cell biology, exemplified here through interrogation of the epithelial-mesenchymal transition (EMT), driving significant improvements in downstream quantitative analyses usually reserved for omics data, including: classification of cell phenotypes; manifold learning of cell phenotype heterogeneity; and pseudotemporal inference of molecular marker dynamics. Introducing data integration concepts from omics to microscopy, ExIF empowers life scientists to use routine 4-plex fluorescence microscopy to quantitatively interrogate complex, multimolecular single-cell processes in a manner that approaches the performance of multiplexed labelling methods whose uptake remains limited.

Heterogeneous cell states and phenotypes arise from complex processes shaped by the levels and subcellular localisations of various types of molecular component, including DNA, RNA, proteins, metabolites and lipids. Over the past decade, many omics modalities have emerged to systematically define molecular component levels with single-cell resolution for (typically) one of these component types at a time across heterogeneous cell populations[1,2]. More recently, computational methods have been developed to integrate different molecular component profiles captured via multiple omics modalities[3,4]. As recently reviewed[5], the most effective of these data integration strategies leverage so-called data anchors, i.e. measured features and/or

cell populations that are common across otherwise independent datasets which guide the quantitative integration process. Particularly in the presence of such data anchors, multimodal data integration can dramatically enrich analyses of complex multimolecular cell states and mechanisms.

Compared to existing omics technologies, fluorescence microscopic imaging remains unparalleled in its ability to provide measurement resolutions (especially sub-cellular spatial and temporal resolution) exceeding those accessible via omics[6]. Yet, despite these advantages in resolving power, almost no emphasis has yet been placed on developing computational approaches that can integrate

[1]School of Biomedical Sciences, Faculty of Medicine and Health, University of New South Wales, Sydney, NSW, Australia. [2]School of Computer Science and Engineering, Faculty of Engineering, University of New South Wales, Sydney, NSW, Australia. [3]Ingham Institute for Applied Medical Research, Liverpool, NSW, Australia. [4]School of Biotechnology and Biomolecular Sciences, Faculty of Science, University of New South Wales, Sydney, NSW, Australia. [5]UNSW Data Science Hub, University of New South Wales, Sydney, NSW, Australia. [6]UNSW Artificial Intelligence Institute, University of New South Wales, Sydney, NSW, Australia. ✉e-mail: john.lock@unsw.edu.au

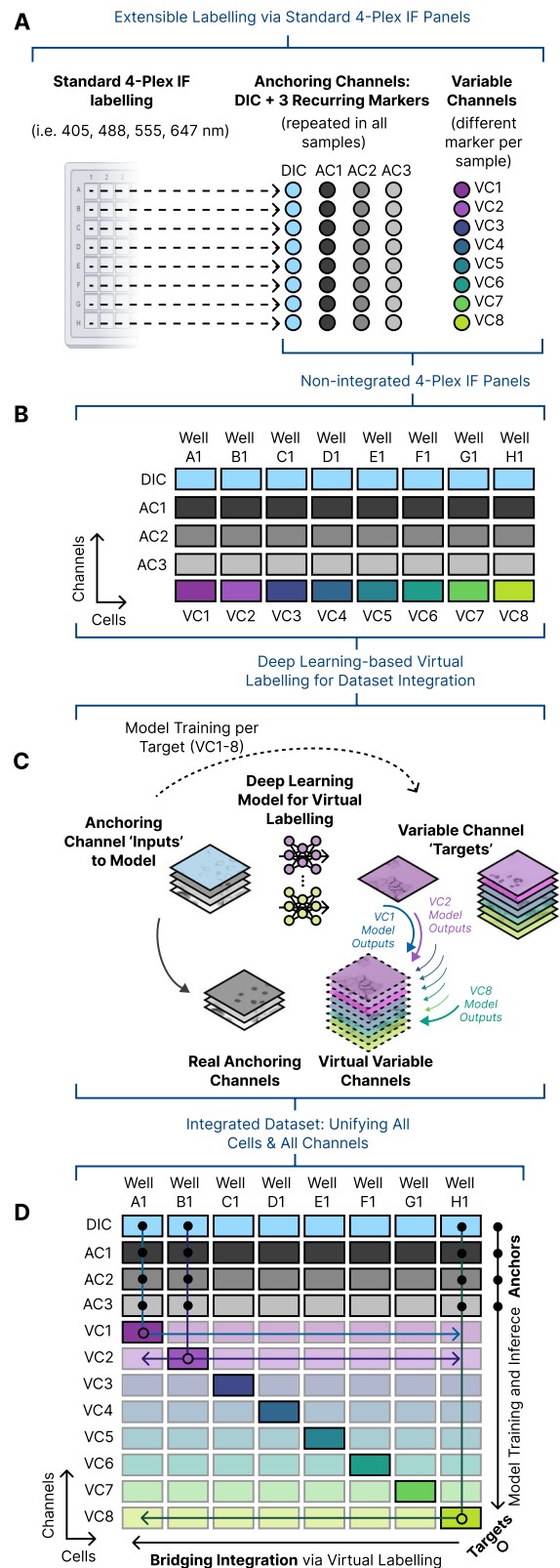

**Fig. 1 | Overview of extensible immunofluorescence (ExIF) workflow. A** Standard 4-plex immunofluorescence (IF) with 'extensible' panel design. Each well yields DIC, three recurring fluorescence markers (also termed 'anchoring channels', AC1-3) and a single variable channel (VC1-8). Markers common to each well (DIC and AC1-3) are used as anchoring channels in subsequent integration steps. **B** Acquisition as per (A) produces datasets where all cells (from all wells) have DIC and AC1-3 images, but distinct variable channels. **C** Integration using deep learning-based virtual labelling is performed. A separate deep learning model is trained for each variable channel using real labelling as training targets from each respective well and anchoring channels as input. Trained models are then used to generate predictive 'virtual channels' for each variable channel in all cells in all wells given the same anchoring channels present in all wells. **D** Overview of the integrated dataset. Each well has real DIC and AC1-3, and a single VC, which are used as inputs and targets, respectively, during model training. Each model is then applied to all remaining wells, generating virtual channels that 'bridge' across all cell populations for each VC, with bold outlines indicating the well used to train the model for a given VC.

limitation remains pervasive despite the emergence of powerful experimental methods enabling high-plexity immunofluorescence (IF) imaging[8–10], whose uptake remains relatively low. Thus, most researchers leveraging the enhanced resolving power of fluorescence imaging are significantly inhibited in their ability to interrogate complex, multimolecular cell processes.

To address this challenge computationally, several virtual labelling approaches have been proposed, although to date their use has overwhelmingly been in the context of cellular imaging within a paradigm of label replacement, i.e. to obviate the need for fluorescence labelling by training deep learning (DL) models to predict a fluorescence marker-of-interest using a label-free input[11–23]. In the context of tissue imaging and digital pathology, efforts have gone further, using deep learning approaches to virtually reconstruct multiplexed data from a smaller number of input channels[24–26] or even from hematoxylin and eosin (H&E) staining[27]. Traditional machine learning methods have also enabled imputation of quantified multiplexed marker features (e.g. expression level per cell), though this approach fails to recapitulate functionally vital information on subcellular marker localisation[28]. These methods have been presented for reconstruction of virtual data matching the marker plexity of the training dataset, meaning that high plexity virtual labelling first requires generation of experimentally multiplexed labelling data. Such approaches thereby implicitly sustain dependence on multiplexed labelling methods that are, as yet, not employed by the majority of fluorescence imaging practitioners.

In contrast, here we have developed Extensible Immunofluorescence (ExIF), a framework that can integrate a theoretically unlimited number of markers from multiple panels of standard (4-plex) IF labelling, thereby increasing biological insights accessible from downstream analyses without requiring experimental multiplexing for prior training of virtual labelling models. This creates a distinct virtual labelling use case, where instead of replacing experimentally multiplexed fluorescent signals with virtual labelling models trained on preceding multiplexed data, standard plexity marker panels are integrated to any arbitrary plexity using virtual labelling models trained within the same dataset. Inspired by multi-omics integration, ExIF thus achieves subcellularly resolved, scalable multimolecular integration underpinned only by routine 4-plex IF methods that are the most widely utilised by researchers globally.

ExIF comprises a mixed experimental and computational framework for generating a multichannel dataset without need for specialised (e.g. spectral) hardware or potentially degradative reagents used in some experimental multiplexing methods[29]. Considering ExIF use when limited to 4-plex imaging (as is typical in standard applications), ExIF instead relies on parallel labelling of multiple standard 4-plex IF panels (each labelling an independent cell population within a single experiment, e.g. in different wells of a multi-well plate) designed to permit extensible labelling (Fig. 1A). Each 4-plex panel contains a

fluorescence imaging data from distinct markers and cell populations to achieve advances in molecular sampling depth and breadth analogous to those in omics fields. Such advances are however needed, because the majority of fluorescence imaging practitioners are limited to concurrent observation of just ~4 molecular components per cell/sample, due to long-standing technical constraints[7] (e.g. spectral bleed-through, lack of antibody species diversity). This low-plexity

mixture of one or more anchoring channels (three depicted herein), which recur in every panel (i.e. across all cell populations), and one or more variable channels, which differ in each additional labelling panel. This extensible labelling allows a theoretically unlimited number (plexity) of molecular markers to be added in parallel for subsequent integration. While this initially gives rise to non-integrated data comprising multiple 4-plex IF image sets spanning various channels and cell populations (Fig. 1B), the anchoring channels are then used as consistent inputs for multiple generative DL models that are each trained to produce virtual labelling[11,12] of a single variable channel (Fig. 1C). Since anchoring channels are (by design) present in all cell populations, all virtual labelling models (one per variable channel) can then be applied in all cell populations, thus integrating every variable channel marker into every cell. ExIF thereby creates a unified data matrix (Fig. 1D) combining all molecular marker channels across all cells, using high-fidelity virtual labelling to mediate a specialised form of mosaic integration[5] that we term bridging integration (reflecting the use of designed data anchors that support DL models to span and connect the dataset). As no independent training set is required, ExIF integration and subsequent interrogation can happen all within the same experiment.

We first validate the ExIF approach by quantifying the improvement in virtual labelling fidelity when using multi-channel inputs (enabled by the extensible labelling scheme) relative to label-free image inputs only (as per typical label-replacement) and by showing that ExIF is compatible with alternative DL architectures for virtual labelling. We then demonstrate the utility of ExIF by showing how a simple implementation can significantly enrich downstream single-cell quantitative analyses. Specifically, in the context of epithelial-mesenchymal (EM) state heterogeneity, we show: **i)** improved machine learning (ML)-based classification of cell state responses to EM-transition (EMT)-modulators (epidermal growth factor, EGF; transforming growth factor-beta one, TGF-β1); **ii)** enhanced manifold learning to map single-cell heterogeneity in EM-phenotypes; and **iii)** more precise inference of EM-state marker dynamics during EMT. In each case, ExIF-integrated datasets far exceed the performance of the standard 4-plex IF datasets from which they are derived, as well as the performance of label replacement strategies based on label-free image inputs, and ultimately approach (though do not yet match) the performance of true experimentally multiplexed labelling of the same experimental conditions.

Overall, ExIF establishes a methodological framework for the computational integration of standard 4-plex fluorescence imaging data, thereby maximising the capacity of such data to illuminate complex cellular processes and responses. ExIF now constitutes a middle-way between accessible but analytically constrained 4-plex IF imaging, and the heightened challenge and analytical value of multiplexed labelling methods. ExIF thereby democratises capacity for high-plexity, imaging-based interrogations of complex, multimolecular cell biology using standard 4-plex IF methods that are among the most commonly used in cellular and biomedical research.

## Results

### Single-cell integration through virtual labelling
Generative DL-enabled virtual labelling mediates the integration of molecular marker channels and cell populations within our ExIF framework (Fig. 1). ExIF is designed to include multiple fluorescence channels that can be used as input to enhance virtual labelling. We hypothesise that this approach should improve prediction fidelity and enable anchored data integration. We therefore begin by comparing virtual labelling quality between label-free image-based predictions and those augmented with one or more fluorescence channels as inputs (schematised in Fig. 2A), testing computational elements of ExIF. We here assess ExIF using a residual vision transformer (ResViT), making use of state-of-the-art self-attention[30]. The ExIF strategy is

nonetheless generalisable for use with any DL model suitable for effective virtual labelling, meaning that the same framework can leverage future advances in DL model architecture that further improve technical virtual labelling performance. To demonstrate the DL model-agnostic nature of the ExIF framework, we show similar quantitative results in terms of virtual marker fidelity, as well as model performance increases with additional input channels, for two other DL architectures that are commonly used in virtual labelling (U-Net[31], cGAN[32]; Supplementary Fig. 1A).

We compare virtual labelling performance across eight alternate molecular marker channels (8-plex, including: α-Tubulin; DNA (DAPI); CoxIV; fibrillarin; GM130; F-actin (phalloidin); β-Catenin; NF-κB-p65) concurrently labelled via the iterative indirect IF imaging (4i) experimental multiplexing protocol[33] in 45,780 DU145 prostate cancer cells (Fig. 2B) and use differential interference contrast (DIC) images as the label-free component. We stress that experimentally multiplexed IF datasets are not required for ExIF, which is explicitly designed to enable high-plexity dataset integration based on standard 4-plex IF data (exemplified in subsequent figures). We here use the experimentally multiplexed dataset only to demonstrate method development and to compare many alternative input image-to-target image combinations in the ExIF framework.

We initially complemented the DIC anchoring channel with a single additional fluorescence marker channel as model inputs (testing all possible alternatives) (Fig. 2B), assessing virtual labelling fidelity using image-level metrics: structured similarity index[34] (SSIM); Pearson's Correlation Coefficient[35] at the pixel level (Pixel-PCC). Note that all comparative results reported here and throughout this study are based on robust 5-fold cross-validation. Most markers predicted using DIC-plus-fluorescence image input combinations achieved statistically significant improvement over DIC image inputs-alone in both SSIM and Pixel-PCC scores (Fig. 2C). The choice of fluorescence input channel did impact the degree of improvement observed, exemplified when predicting β-Catenin or GM130 virtual labels, with F-actin and α-Tubulin respectively improving image metric scores more than any other marker when added to DIC (Fig. 2D and Supplementary Fig. 1B). Extending this multichannel input strategy to the limit of our 8-plex dataset, we built ResViT (Fig. 2C) as well as U-Net and cGAN models (Supplementary Fig. 1A) incorporating DIC plus seven fluorescence marker channels as input to predict the eighth marker not included as input (in all possible combinations). Virtual image fidelity metrics further improved with significance across most target markers, though some gains were only incremental, especially for the SSIM metric. Nonetheless, the consistent increases in virtual labelling fidelity achieved through use of multiple image inputs validates the premise for the ExIF panel design (adding up to 3 fluorescence channel inputs to DIC) to improve virtual labelling and thereby enhance marker integration.

### Applying extensible immunofluorescence to investigate epithelial-mesenchymal cell state plasticity
In this section, we demonstrate the end-to-end use of ExIF in a practical use case based only on standard 4-plex image data. This demonstration includes several treatment conditions and single-cell analysis of the data produced, showing how ExIF multiplexed data can achieve biological resolution significantly beyond that of standard 4-plex IF image data that is used as input. This demonstration is applied in the context of epithelial-mesenchymal (EM) plasticity, a phenomenon facilitating cancer progression by enabling tumour cells to switch between epithelial and mesenchymal phenotypes[36,37]. Epithelial-mesenchymal transition (EMT) is known to be a complex multi-molecular process, involving repression of epithelial markers and upregulation of mesenchymal markers enabling enhanced cell migration, invasion and thus metastatic progression[38]. To assess the utility of ExIF for interrogating EM cell state plasticity, we amplified EM state

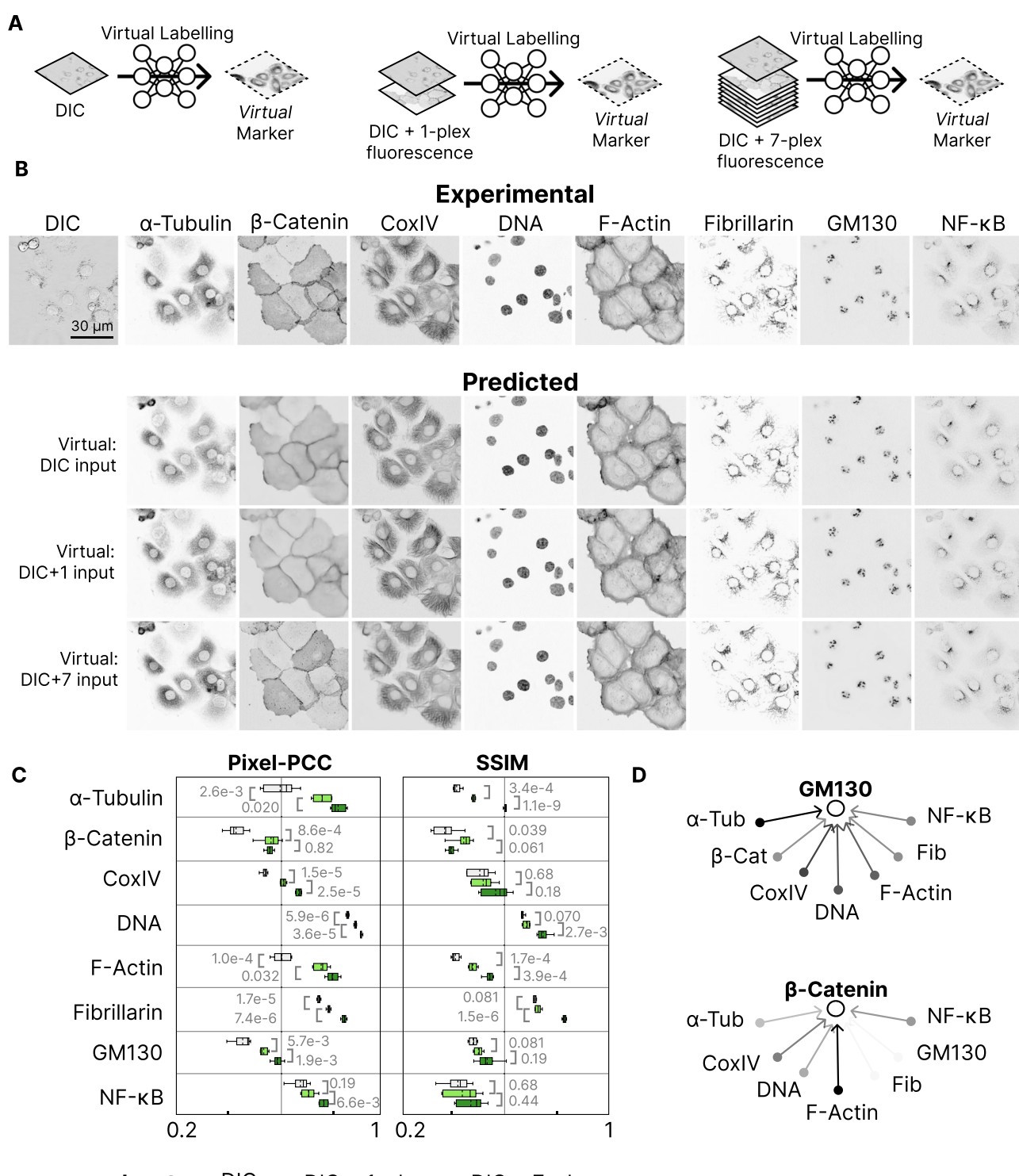

**Fig. 2 | Label-free and multi-modal virtual labelling comparisons. A** Schemas depicting virtual markers (dashed image outlines) generated by virtual labelling using inputs including DIC images only (Left), DIC images plus a single (1-plex) fluorescence marker (Centre), or DIC images plus seven (7-plex) fluorescence markers (Right). **B** Experimental markers/channels as titled used for training and evaluation (Top); predicted using ResViT model with various inputs (Bottom). **C** Median image metric scores across 5 cross-validated folds. Different input schemes shown in colour with only the best-performing pair for DIC + 1-plex shown. Remaining DIC + 1-plex performance shown in Supplementary Fig. 1B. Significance testing reflects Welsh's two-sided t-test. All box plots depict data distributions as follows: dashed centre line, median; solid centre line, mean; box limits, upper and lower quartiles; whiskers, 1.5x interquartile range. **D** Arrow weights indicate improvement of Pixel-PCC (Pearson's correlation coefficient; white-to-black indicates low-to-high improvement) when adding different fluorescence channels to DIC as ResViT inputs for virtual labelling of GM130 (top) or β-Catenin (bottom). Source data provided as detailed in data availability.

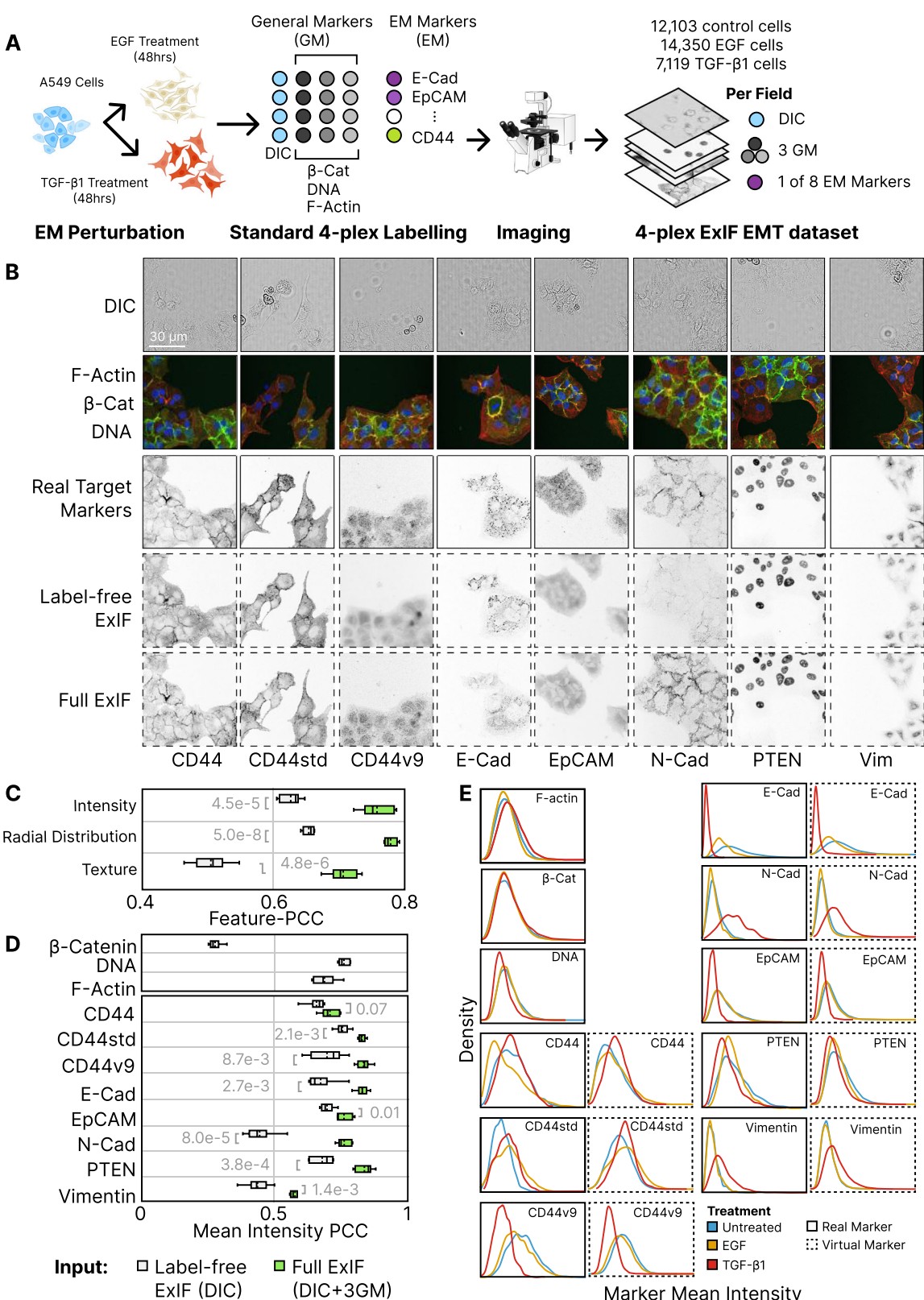

diversity in A549 lung cancer cells by comparing control cells to those treated for 48 h with either EGF (0.1 µg/mL) or TGF-β1 (0.01 µg/mL); growth factors known to weakly or strongly (respectively) drive EMT in A549 cells[39]. To readout EM state per cell, we labelled and imaged eight different 4-plex IF panels across each treatment condition, following the extensible labelling schema schematised in Fig. 1A & B. Specifically, each panel consists of recurring general markers (DNA, F-Actin and β-

Catenin), and a single variable EM state marker (E-Cadherin or EpCAM or N-Cadherin or PTEN or Vimentin or CD44total or CD44std or CD44v9[40–45] (Fig. 3A). We then performed data integration via virtual labelling, using the recurring general markers and DIC as inputs to ResViT virtual labelling models to predict labelling for variable (non-recurring) EM state markers, in line with the virtual labelling and dataset integration schema depicted in Fig. 1C & D. Note that one DL

**Fig. 3 | Experimental modulation of epithelial-mesenchymal (EM) state & optimised virtual labelling of EM markers. A** Experimental design for ExIF workflow. A549 cells were labelled using three 'general markers', and a 'variable' Epithelial-Mesenchymal (EM) state marker that differed in each well. Imaging thus incorporated label-free DIC, the three general markers and a single EM marker, per field. Microscope icon by DBCLS licensed under CC-BY 4.0 Unported, Cell icons created in BioRender Lock, J. (2025) https://BioRender.com/ftfhrx5. **B** Real channels used as input (Top two rows). Target variable markers as labelled below used for training and evaluation (3rd row). ResViT predictions using label-free or multichannel inputs (4th and 5th rows). Images contrast-adjusted equivalently per marker for display only. **C** 5-fold cross-validated single-cell feature-level PCC (Pearson's correlation coefficient) scores per feature category across all eight EM markers. **D** 5-fold cross-validated single-cell feature-level PCC scores for mean cell intensity

per marker. Note: β-catenin, DNA and F-actin predicted using DIC inputs only as these are the only markers common to all cells (avoiding self-prediction). **E** Comparison of marker mean intensity distributions between real and virtual labels. Normalised histograms of cell mean intensities per marker. Histograms with solid borders reflect real marker distributions. Histograms with dashed borders reflect virtual label distributions generated via ResViT models using DIC plus the 3 general markers as input. Note: As general markers, β-catenin, DNA and F-actin did not undergo virtual labelling (avoiding self-prediction). All box plots depict data distributions as follows: dashed centre line, median; solid centre line, mean; box limits, upper and lower quartiles; whiskers, 1.5x interquartile range. All significance reflects Welsh's two-sided t-testing. Source data provided as detailed in data availability.

model is trained for each (variable) EM marker. Training was performed using images from three wells per variable marker (one well per treatment condition) to finetune models that were pretrained using ImageNet21K. We highlight the role played by the recurring general markers in the ExIF context, which provide consistent virtual labelling inputs and act as common anchors for multimolecular data integration. This unified 4-plex ExIF EMT dataset thus includes all cells characterised by all 11 fluorescence markers (3 real general markers & 8 virtual EM state markers). Although ExIF is designed to enable utilisation of multiple input channels to improve integration performance, we note that some research applications are suited to a label-free only approach (e.g. long-term live cell imaging, sensitive diagnostic sample imaging etc). Thus, in addition to assessing integration performance using full ExIF (using general fluorescence markers and DIC as input to predict each of the 8 EM-state markers), we also assessed using DIC as the only input; termed label-free ExIF (Fig. 3B).

Finally, because ExIF aims to maximise integrated biological insights using standard 4-plex data inputs, it is important to compare ExIF performance (for downstream biological analyses) to true experimental multiplexing, since this remains—where feasible—the gold standard for imaging-based multimolecular analysis. To directly compare ExIF versus real experimental multiplexing performance, we therefore generated an analogous experimentally multiplexed EMT dataset again using A549 cells (137,893) with identical treatment conditions (as above), labelling all cells in all conditions with a panel of 13 markers (plus DIC imaging), including 3 fluorescence markers designated (for the purposes of ExIF simulation) as general channels (DNA (DAPI), β-catenin, Actin) and 10 EM state markers designated as variable channels (CD44, E-cadherin, EpCAM, β-tubulin, N-cadherin, pSMAD2/3, pan-Cytokeratin, Vimentin, ZEB1, zonular occludins (ZO)-1) (Supplementary Fig. 2A). Virtual labelling fidelity and biological analysis performance data from this experimentally multiplexed EMT dataset, including emulation of 4-plex ExIF performance through subsampling of the full multiplexed panel, is presented across Supplementary Fig. 2, mimicking key results presented for the true 4-plex ExIF EMT dataset analysis across Figs. 3–5. Notably, as the experimental multiplexing was in this case performed using the Cyclic IF protocol[9] that relies on fluorescently conjugated primary antibodies, the experimentally multiplexed EMT marker panel is similar but could not be identically matched to the 4-plex EMT marker suite presented in the 4-plex ExIF EMT exemplar, which utilised unconjugated antibodies (plus DAPI (DNA) and phalloidin (F-actin)) optimised for standard 4-plex indirect IF.

To analyse the performance of ExIF-mediated virtual labelling and dataset integration with an emphasis on single-cell data as the key point of biological focus, we used CellProfiler-based cell segmentation and quantitative feature measurement to extract 17 morphological features (cell size, shape, neighbour-contact etc) plus 142 features per marker per cell, as commonly performed in cell phenotype profiling (features listed in Supplementary Table 1). Data spanned 12,103 control cells, 14,350 EGF-treated cells and 7119 TGF-β1-treated cells in the

4-plex ExIF EMT dataset (Figs. 3–5), and 42,304 control cells, 61,497 EGF-treated cells and 34,092 TGF-β1-treated cells in the experimentally multiplexed EMT dataset (Supplementary Fig. 2). For each quantitative feature, we calculated the Pearson's correlation coefficient (metric herein termed Feature-PCC) between ground-truth experimentally labelled and virtually labelled marker channels as a measure of the fidelity of the biological cell profile.

For the 4-plex ExIF EMT dataset, we found significantly higher Feature-PCC values between real and virtual labels when using full-ExIF (all inputs utilised; DIC + 3 general markers) (Fig. 3C; Feature-PCC values per feature category, per marker shown in Supplementary Fig. 3A), consistent with our previous results showing the same trends at the level of image quality metrics. In particular, multichannel inputs achieved high feature correlations (~0.7–0.9 PCC) for per cell mean intensity (a crucial feature used as a proxy for protein expression), though vimentin showed weaker performance; <0.6 PCC (Fig. 3D). The same performance improvements were observed in the experimentally multiplexed EMT dataset (Supplementary Fig. 2B).

Returning to the 4-plex ExIF EMT dataset, we compared real and virtual EM state marker mean intensity distributions to examine differences induced by growth factor treatments (Fig. 3E), and the degree to which these were recapitulated in virtual signals. In real signals, we found that control and EGF conditions had relatively similar distributions, whereas TGF-β1 induced more dramatic changes. Mesenchymal markers N-cadherin and Vimentin were especially strongly expressed after TGF-β1 treatment, while epithelial markers CD44v9, E-Cadherin, and EpCAM were strongly suppressed. These effects are consistent with established EMT biology[40–45]. Notably, EM marker distributions resulting from ExIF-mediated dataset integration closely reflected treatment-induced effects seen with real labels (i.e. those derived from the subset of cells experimentally labelled with each EM marker in this dataset). Thus, ExIF achieves strong performance across various EM state markers, retaining biologically relevant information that captures key responses to different EM-state perturbations, including inherent inter-cellular response-heterogeneity (signal distributions).

## Classifying cell treatment condition using ExIF integrated data
Using the 4-plex ExIF EMT dataset, we next assessed whether ExIF-mediated data integration would enhance downstream analyses commonly used to interrogate complex cell biology, including in the contexts of omic and multi-omic analyses. We first tested accuracies for machine learning classification of treatment conditions (control vs EGF vs TGF-β1) according to the scheme depicted in Fig. 4A. Using standardized and scaled single-cell (CellProfiler-derived) features from raw 4-plex images and ExIF-integrated datasets based on various image input and real / virtual image output combinations (defined in left table, Fig. 4B), we performed principal components analysis (PCA) before using derived principal components as input for support-vector machine (SVM)-based classification of cell treatment conditions. This approach was repeated for each cell population labelled with a different EM state marker, and classification results were evaluated via F1-

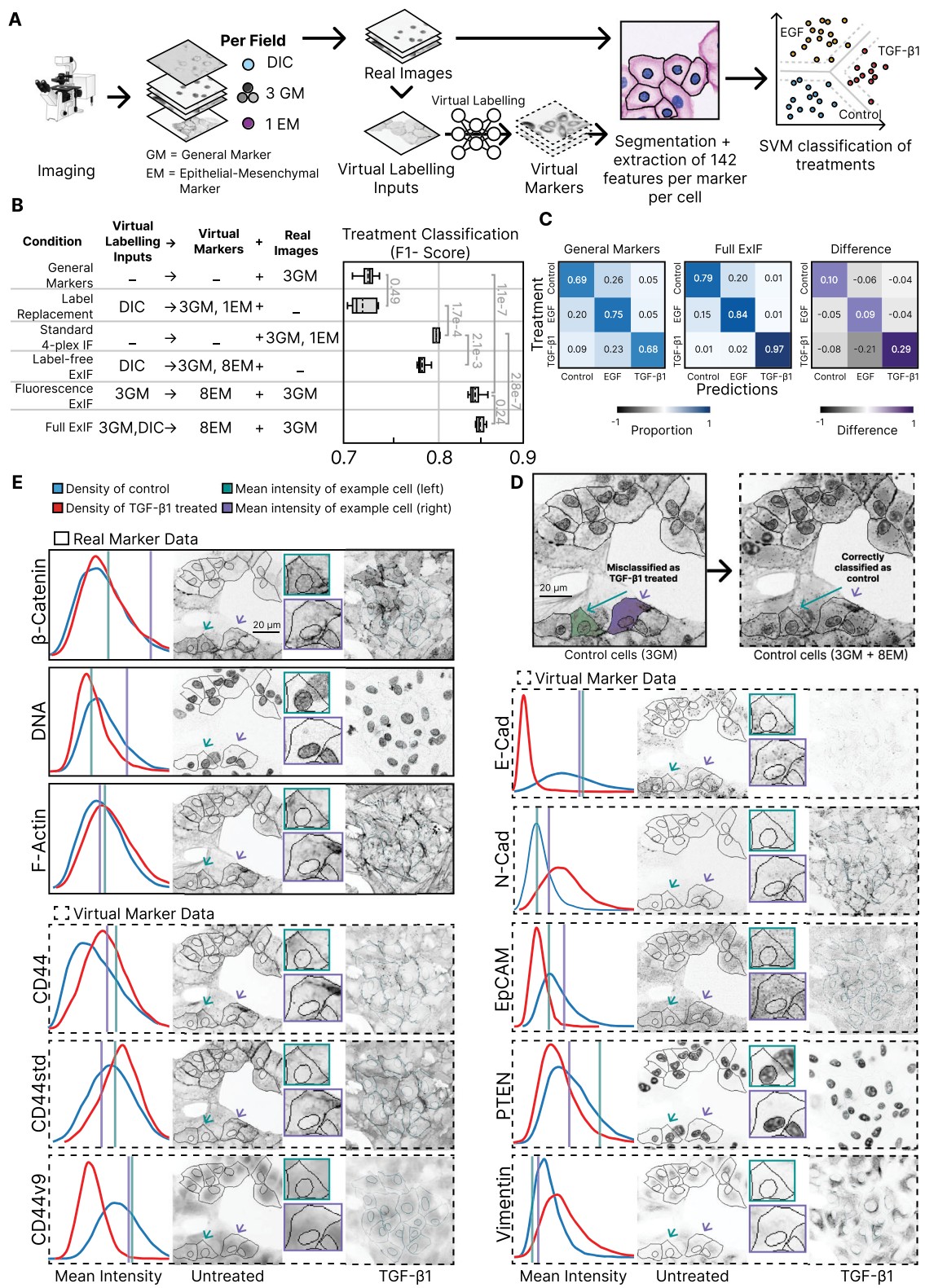

scores, comparing different datasets in aggregate (results in right boxplot, Fig. 4B).

We found that full ExIF achieved significantly better classification of treatment conditions (F1 ~ 0.85) compared to standard 4-plex IF (F1 ~ 0.8). We also noted that ExIF anchored using only the three general fluorescence channels (no DIC) retained classification performance (F1 ~ 0.84) that was far better than when using only features directly measured from the three general fluorescence anchoring channels (F1 < 0.75). This implies that ExIF integration of EM-state markers (via virtual labelling) adds significant discriminatory value (regarding treatment class) beyond that directly accessed from the general markers that form the basis of integration. Notably, we included CellProfiler-measured spatial and cell morphological features (e.g. cell-cell contact levels and intercellular distances; cell size, shape)

**Fig. 4 | ExIF improves phenotypic classification of experimental treatments modulating epithelial-mesenchymal state in comparison to standard IF.**
**A** Workflow comparing cell phenotype classification using different real/virtual label combinations. Fields imaged for DIC, 3 'general markers' (GM) and a single epithelial-mesenchymal marker (EM) (described in Fig. 3) and used for virtual labelling and for cell segmentation/feature extraction. Combinations of features from real/virtual markers underpinned support vector machine (SVM) classification of phenotypes from control, EGF, and TGF-β1 conditions. Microscope icon by DBCLS licensed under CC-BY 4.0 Unported. **B** 5-fold cross-validated classification accuracy (F1-score) using: different virtual labelling inputs (table, left column); features from virtual markers (table, middle column); and features from real images (table, right column). General markers, single variable marker, and all 8 variable markers respectively denoted by 3GM, 1EM, 8EM. Box plots depict data distributions as follows: dashed centre line, median; solid centre line, mean; box limits, upper and lower quartiles; whiskers, 1.5x interquartile range. Significance reflects Welsh's two-sided t-testing. **C** Confusion matrices for classification using real

general markers (left; as per first row in (**B**)); or Full ExIF (centre, as per last row in (**B**)). Differences in performance across these confusion matrices (right).
**D** Example control cells (green, purple) misclassified as TGF-β1-treated when using general markers (as per first row in (**B**)) and corrected by hybrid multiplexing using real general markers and 8 virtual EM-state markers (as per last row in (**B**)) (markers as labelled combined in grey). **E** Comparison of real (solid boxes) or virtual (dashed boxes) mean intensity distributions (normalised histograms) for cells from control (blue) or TGF-β1-treated (red) conditions. Mean intensity values of the control cells denoted (vertical lines as per (**D**)). For these cells, real general marker intensities provide little distinction between treatments. By contrast, several virtual EM labels strongly distinguish conditions, correcting classification of the exemplar cells. Left images show real/virtual labels from the exemplar field (from (**D**)), cropped images (centre) show exemplar cells. For comparison, right images show an exemplary TGF-β1-treated field for the same indicated real/virtual marker. Source data provided as detailed in data availability.

capturing local cellular context in all models (Supplementary Table 1), to avoid an imbalanced performance comparison resulting from the potential encoding of spatial information by ExIF models, which would (otherwise) not be available to the SVM classification models containing real image data inputs-only. Thus, the enhanced classification performance of the fluorescence ExIF data compared to the general markers data appears to be due to the integration of the 8 additional virtual EM markers, mediated by ExIF. Label-free ExIF, which virtually labels all 11 fluorescence markers from this 4-plex ExIF EMT dataset also significantly improved classification beyond that achieved using the three (real) general markers. Label-free ExIF achieves similar EM-state discriminatory power with label-free images-only as can be achieved using features from a standard 4-plex IF experiment that includes at least one explicit EM-state marker. By comparison, we note that virtual labelling of only the original 4-plex panels using DIC-only-inputs – an approach that mimics the prevailing label-replacement paradigm for virtual labelling – only approximated classification results achieved using features measured from the three general markers.

Considering the experimentally multiplexed EMT dataset, emulation (by subsetted training) of 4-plex ExIF-integrated data (F1 ~ 0.89) in comparison to true experimentally multiplexed data (F1 ~ 0.96) showed that ExIF achieves an accuracy approaching but, as expected, not quite matching true experimental multiplexing data (Supplementary Fig. 2C). Nonetheless, the various forms of ExIF all greatly outperformed standard 4-plex (F1 ~ 0.81) and 3-plex (general markers; F1 ~ 0.73) performance for treatment classification.

Interestingly, in the 4-plex ExIF EMT dataset, classification using only features from the three general markers most often misclassified control and TGF-β1 cells as EGF-treated (Fig. 4C), while full ExIF integration reduced misclassifications across all classes, especially improving differentiation of TGF-β1 and EGF conditions. To exemplify how these performance improvements arise, we show how the addition of virtual markers corrects two control cells misclassified as TGF-β1-treated (Fig. 4D). Comparing mean intensity distributions, we see no clear separation between control (blue) and TGF-β1 conditions (red) when first considering the real general markers only, although TGF-β1-treated cells have somewhat lower mean DNA intensities (Fig. 4E). Given these minimal general marker differences between control and TGF-β1 conditions, the two misclassified control cells (mean intensities marked by vertical lines - green or purple) could belong to either treatment population. Indeed, the real general marker labelling of these control cells looks similar to exemplar TGF-β1-treated cells. In contrast, EM-state marker integration, especially of CD44v9, E-Cadherin, N-Cadherin, EpCAM, and Vimentin, defines population distributions that more effectively discern control and TGF-β1 conditions. Accordingly, the two example cells can be identified as strongly exhibiting control phenotypes when considering the

combination of all virtual markers, clearly and explicitly exemplifying how ExIF integration enhances phenotype discrimination and classification capacity.

## Mapping EM-state heterogeneity and marker dynamics using ExIF data integration

Moving beyond classification of the discrete treatment conditions, we next examined whether ExIF integration could enhance the quantitative mapping of cell phenotype heterogeneity in the EM spectrum. To this end, we applied several unsupervised manifold learning methods to our quantitative single-cell feature datasets. We present manifolds constructed with PHATE, a manifold-embedding technique that seeks to capture underlying dynamic trajectories in multivariate data[46] (Fig. 5), as well as manifolds constructed with T-SNE[47] and UMAP[48] (Supplementary Fig. 3B). Importantly, we here excluded spatial context and cell morphology features (relating to cell-cell contact, cell neighbour proximity, cell size, cell shape) from construction of these and all subsequent manifolds, because cell spatial context and morphology are linked to EM state, and thus these features would undermine capacity to assess the construction of EM-state-sensitive manifolds leveraging ExIF-integrated molecular marker data.

Using the 4-plex ExIF EMT dataset, we first constructed a Standard manifold using CellProfiler-derived quantitative features from the three recurring general markers, since these are the only real markers that are present in all cells (Fig. 5A). This Standard manifold shows only a very a limited capacity to resolve the three experimental conditions. We next constructed manifolds via label-free ExIF integration of all 11 markers using DIC as the only DL model input (Fig. 5B). We found that this label-free ExIF integrated data maps EM-state heterogeneity more effectively than standard IF data, with improved treatment condition-separation and manifold extension. Lastly, we generated full ExIF manifolds (using general markers plus DIC as DL model inputs), integrating all eight EM-state markers (Fig. 5C). Features measured from this optimally integrated dataset greatly improved delineation of treatment conditions in the PHATE manifold, emphasising differences between the mesenchymal states induced by EGF versus TGF-β1[49–51], as well as defining a more structured and extended manifold profile than the other manifold versions. T-SNE and UMAP manifolds confirm similar tendencies (Supplementary Fig. 3B).

Modelling of pseudotime[52] within the PHATE-embedded manifolds further supports improved cogency of the full ExIF manifold, which defines a trajectory capturing progression from cell states enriched for the control condition to cell states enriched for EGF then TGF-β1 treatments. Representative cell images from milestones (1-6) along the full ExIF manifold pseudotime trajectory indicate cell morphological trends aligned with EMT (Fig. 5D). Indeed, colour-coding the full ExIF manifold by real (CellProfiler-measured) morphological and cell-cell contact values (not used in manifold construction) shows

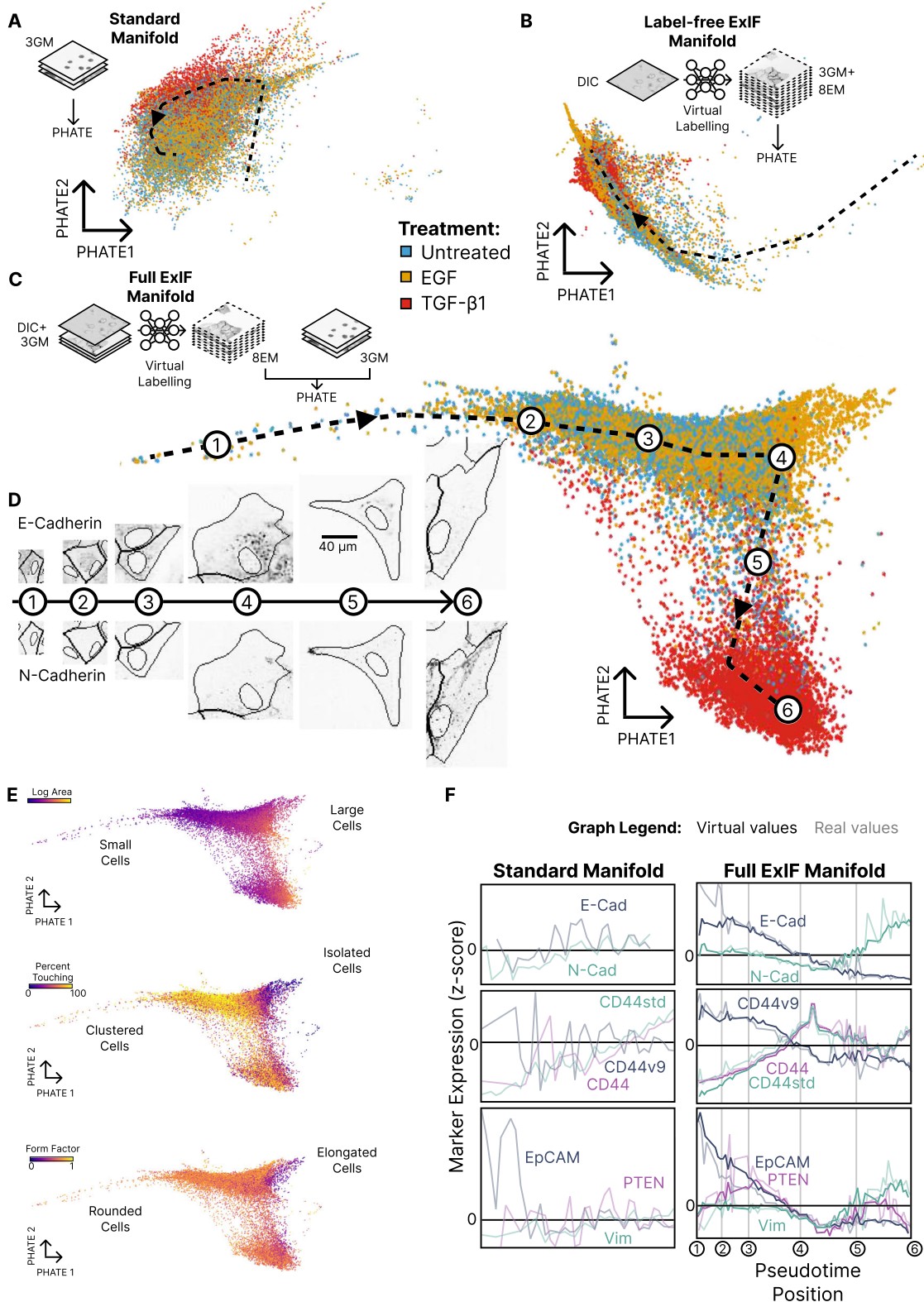

consistent quantitative trends aligned with expected EMT dynamics, i.e. increasing cell area, increasing cell protrusivity and reduced cell-cell contact (Fig. 5E). Pseudotime analyses further delineated individual EM marker-level (mean cell intensity) variations along each inferred trajectory. These variations are unstructured and noisy along the pseudotime trajectory of the Standard manifold compared to clear EM marker trends and signals spanning the pseudotime trajectory of full ExIF manifold (Fig. 5F); again supporting improved manifold

characteristics achieved via the optimal full ExIF dataset integration. Notably, the displayed EM marker dynamics derived from the Standard manifold (Fig. 5F, left) reflect real marker values from the cell sub-populations experimentally labelled for each variable marker. EM marker dynamics from the full ExIF manifold (Fig. 5F, right) depict either: real marker dynamics (weak line colours; from the subset of cells experimentally labelled for each marker); or virtual marker dynamics (strong line colours; from all cells). Comparisons of real

**Fig. 5 | Computational multiplexing via virtual labelling enhances manifold-learning of epithelial-mesenchymal (EM) phenotype heterogeneity and inference EM marker dynamics. A** 'Standard' phenotypic manifold constructed using features extracted from 3 real common markers only. Manifolds in **A**–**C** reflect PHATE embeddings for 33,572 cells (12,103 untreated (blue); 14,350 EGF-treated (orange); 7,119 TGF-β1-treated (red)) and do not incorporate morphological data (size, shape, cell-cell contact etc); comparable t-SNE and UMAP embeddings in Supplementary Fig. 3B. Black dashed lines in **A**–**C** reflect pseudotime trajectories inferred using scFates. **B** 'Label-free ExIF' phenotypic manifold constructed using features extracted from 11 virtual labels (3 common plus 8 EM markers) predicted by ResViT using label-free (DIC) input. **C** 'Full-ExIF phenotypic manifold constructed using features extracted from the three real general markers (described in Fig. 3) and eight virtual EM markers predicted by ResViT using the three general markers and DIC as input. The pseudotime trajectory in **C** includes 'milestones' (1-6) exemplified by representative images (**D**). Changes in morphology, cell-cell contact, and virtual EM labels, including E-cadherin (top row; epithelial marker; expression decreases from milestone 1 to 6) and N-cadherin (bottom row; mesenchymal marker; expression increases from milestone 1 to 6). **E** The Full-ExIF manifold from **C** is colour-coded by real morphological and spatial context features extracted from general marker-enabled cell segmentation confirming morphological changes congruent with an epithelial-to-mesenchymal transition (EMT)-like pseudotime trajectory. **F** Z-score normalised average EM marker intensities (y-axes) indicate EM marker dynamics relative to pseudotime (x-axes) defined in either the Standard (left: as in **A**) or Full ExIF (right: as in **C**) phenotypic manifolds. Real EM marker dynamics (from the subset of cells experimentally labelled for each EM marker, colour coded as indicated) inferred from the Standard manifold pseudotime trajectory show noisy expression-level variations with little relation to known EMT biology (Left). Virtual EM marker dynamics (strong lines) inferred from Full ExIF manifold pseudotime trajectory match known EMT biology and real morphological changes (in **E**) (Right). The virtual marker dynamics also closely match real marker dynamics (faded lines) from the subset of cells experimentally labelled for each EM marker. Source data provided as detailed in data availability.

versus virtual marker trends reveal strong correspondence for each marker, indicating that dynamical trends inferred using the virtual data used for dataset integration correspond to ground-truth marker dynamics, though the virtual data trends are substantially less noisy.

Focusing on the specific expression-level dynamics inferred for each EM marker in the optimally performing full ExIF manifold, we see trends aligned with established molecular changes during transition from epithelial to mesenchymal state. E-cadherin expression declines while N-cadherin levels ultimately elevate[42]. The epithelial state-associated CD44v9 isoform is suppressed, counterbalanced by increases in total CD44 and in the mesenchymal state-associated CD44std isoform[53]. Tumour-suppressor PTEN expression declines while vimentin levels peak late in the trajectory[43,44]. As with E-cadherin, epithelial marker EpCAM also declines substantially[41]. By contrast, these biologically informative trends are difficult or impossible to discern from pseudotime analysis of the Standard manifold built using data from the common general (anchoring) marker channels-alone.

Considering results from the experimentally multiplexed EMT dataset, emulation (by subsetted training) of 4-plex ExIF-integrated data achieves a PHATE manifold structure that closely matches that from the real experimentally multiplexed dataset, though the latter has somewhat higher treatment condition-resolution. By contrast, the Standard manifold structure is poorly resolved and gives rise to a pseudotime trajectory with limited interpretability (Supplementary Fig. 2D and E). Accordingly, the inference of selected EM marker dynamics based on the Standard manifold shows little correspondence with the dynamics inferred from the manifolds and trajectories based on emulated ExIF and real multiplexing, whose fundamental trends are robust and also highly analogous to each other. Once again, strong correspondence is also observed between the inferred real EM marker dynamics and virtual EM marker dynamics within the ExIF manifold.

Taken together, we find that ExIF integration – even when using only label-free (DIC) inputs – can achieve analytical outcomes that outperform interrogation of standard IF alone or the 3 general marker data, despite in some comparisons being based on that same exact underlying image data. Moreover, our full ExIF approach, which achieves high-fidelity integration to create high-plexity fluorescence imaging data, enables powerful multimolecular analyses of discrete cell states, continuous phenotypic heterogeneity and (pseudo-temporal) marker dynamics. Achievable with just a single experiment comprising any desired number of standard 4-plex IF labelling conditions (and hence any arbitrary number of integrateable markers), these results approach, though do not yet match, the performance of actual experimentally multiplexed data - whilst maintaining accessibility to the large number of fluorescence imaging practitioners that routinely use standard 4-plex fluorescence imaging techniques.

## Discussion

Compared to the single-cell omics domain[1,2] where computational methods for data integration have rapidly proliferated, equivalent computational integration methods remain entirely lacking for fluorescence imaging-based research applications, where the benefits of heightened resolution[6] have long been counterbalanced in the context of multimolecular biological analyses by severe restrictions on molecular marker plexity. Though several experimental methods permitting high-plexity fluorescence imaging have emerged[10,33,54], their uptake remains low compared to the number of laboratories routinely generating 4-plex IF data. Accordingly, to democratise capacity for high-plexity fluorescence imaging analyses we have developed the Extensible Immunofluorescence (ExIF) framework, combining concepts from multimodal omics data integration with generative deep learning-enabled virtual labelling. ExIF unifies carefully designed extensible labelling panels of otherwise routine 4-plex IF images into theoretically unlimited-plexity datasets using virtual labelling to mediate computational data integration.

This represents a significant advance in the framing of virtual labelling utility, where to date, emphasis has been on label-replacement, which replaces real labels with virtual labels to maintain specific molecular insights. Such label-replacement is enacted either in the absence of experimental fluorescence labelling (i.e. using label-free image inputs), whereby accuracy and plexity remain limited[11–22]; or more recently, by replacing multiplexed labelling using strategies that nonetheless rely on prior model training using experimentally multiplexed datasets[24–26]. Now, within one experiment, ExIF integrates standard 4-plex IF data to produce virtually multiplexed datasets with theoretically unlimited potential plexity. We have demonstrated the end-to-end application of ExIF using a strictly 4-plex ExIF EMT dataset as well as an experimentally multiplexed (13-plex) EMT dataset, which allows direct comparison of emulated 4-plex ExIF integration versus ground-truth experimentally multiplexed data. Consistently, ExIF integrates cells and markers with high fidelity to significantly enrich downstream interrogation of both discrete (via classification) and continuous (via manifold learning) cell phenotype heterogeneity, achieving levels of analytical performance beyond that of the underlying 4-plex input data, and approaching (though not yet matching) that of experimentally multiplexed data. ExIF thus accessibly enables quantitative, single-cell, imaging-based analyses of multi-molecular cellular states and mechanisms using only routine 4-plex imaging data to nonetheless interrogate highly complex cellular biology.

Though until now lacking translation to the domain of fluorescence imaging, growing interest in integration for omics data[3,4] has led to a conceptual schema defining several types of data integration strategy. These include so-called diagonal, horizontal, vertical and mosaic integration strategies that are defined by the presence and

nature of the data anchors (recurring cell populations and/or molecular features) that serve to connect disparate datasets[5]. Briefly, diagonal integration is necessary (though challenging) where no common data anchors link independent datasets, whereas common molecular features (horizontal integration; i.e. equivalent feature-sets measured across multiple cell populations) or cell populations (vertical integration; i.e. multiple feature-sets measured in one cell population) anchor integration in other contexts. Finally, mosaic integration relies on partial overlap (often random) between cell populations and/or measured features enabling data-driven connections between all data subsets, wherein any pair of data subsets may be linked by different anchors. Since ExIF performs both cell-wise and feature-wise integration, it is akin to mosaic integration, yet it differs in that our (though easily reproduced) extensible labelling strategy allows us to design the anchoring features that are common across cell populations, giving opportunities to optimise their integrative performance while also removing the random element of overlap that can undermine mosaic integration performance. Given the structured nature of the anchors that support DL models to connect across the total span of the dataset, we refer to ExIF as a bridging integration strategy. This bridging approach exemplifies how careful experimental design can aid in the formulation of systematic integration approaches and bring a data integration strategy to the domain of fluorescence imaging.

ExIF-mediated improvements in analytical performance are particularly noteworthy in several cases. In our analysis of EM states, comparing treatment classification performance between the general markers (3GM) condition and the fluorescence ExIF condition (which uses the same 3 general markers as anchors for data integration) reveals a remarkable increase in classification accuracy due to virtual labelling-enabled data integration. This effect is observed in both of the (4-plex and experimentally multiplexed) EMT datasets. Crucially, this is despite both analyses ultimately relying on identical underlying image data (the three general markers). Similar analytical performance improvements were observed when comparing standard manifolds (based on the three general markers) to full ExIF manifolds—with the major differences in input to these manifolds being the integration of all virtual EM-state markers (into the full ExIF manifold). Together, these results suggest that data integration through the virtual labelling process functions as a form of feature engineering[55] that surfaces latent information in the general anchoring channels and makes this more accessible to the finite machine learning models used for either classification (SVM) or manifold learning (PHATE, t-SNE, UMAP) – thereby improving their performance.

When considering the design of experiments employing the ExIF framework, we note differences in virtual labelling performance (and thus dataset integration) depending on the selection of anchoring markers. While we have chosen morphological (DNA, actin) and EM-state adjacent (β-catenin) markers to act as anchors, our results suggest there is potential to optimise anchoring channel selection to improve the fidelity of targeted virtual labels. At the same time, we also saw that ExIF integration anchored by the label-free (DIC) channel achieved analytical performance matching that of standard 4-plex IF. This indicates significant potential for such high-plex virtual label integration to enhance cell phenotype analyses even in purely label-free imaging applications such as ultra-high-throughput screening, long-term live cell imaging, or minimally invasive imaging of sensitive clinical samples[56].

In summary, we have combined extensible labelling experimental design with application of virtual labelling to establish the Extensible Immunofluorescence (ExIF) framework. In contrast to previous virtual labelling, ExIF is end-to-end, relying only on a single experiment/dataset for training and prediction, integrating markers present into a multiplexed dataset beyond the plexity restrictions of the original data. Distinctly, ExIF is not limited in plexity[11–22], nor does it require initial experimental multiplexing for model training[24–26]. Here,

demonstrated using standard 4-plex IF, ExIF enables simultaneous integration of both cells and molecular markers to generate unified datasets with high-fidelity to experimental labelling. We have demonstrated significant improvements compared to standard IF in the capacity to perform several downstream quantitative single-cell, multimolecular analyses. Thus, the ExIF framework achieves a step-change in the utility of standard IF as a method for quantitative single-cell analyses of complex, multimolecular biology, which approaches the levels of analytical performance previously only accessible with multiplexed labelling methods. This democratises the powerful benefits of high-plexity fluorescence imaging for the thousands of laboratories within which standard IF methods are already routine.

## Methods
### Cell culture and imaging
**Cell culture.** DU145 and A549 cells were cultured in Minimum Essential Medium (MEM, Thermo Fisher Scientific) and Dulbecco's Modified Eagle Medium (DMEM, Thermo Fisher Scientific), respectively. Both mediums were supplemented with 10% Fetal Bovine Serum (FBS, Thermo Fisher Scientific), 4 mM L-glutamine (Thermo Fisher Scientific) and 23.4 mM 4-(2-hydroxyethyl)-1-piperazineethanesulfonic acid (HEPES, Thermo Fisher Scientific).

Cells were passaged two to three times per week and used within passage number 3 and 20. Cells were cultured at 37 °C with 5% $CO_2$ to 85% confluence before harvesting with 1x phosphate-buffered saline (PBS) and 0.05% Trypsin-EDTA (Thermo Fisher Scientific) incubation for 3-5 min.

**Iterative indirect immunofluorescence imaging (4i).** For 4i multiplexing experiments, 10,000 DU145 cells were seeded into 96-well glass bottom plates (Cellvis) and 4i was performed[8,33]. Cells were fixed with 4% paraformaldehyde (PFA, Electron Microscopy Sciences) in PBS for 20 min and permeabilised with 0.1% Triton X-100 (Sigma-Aldrich)/PBS for 10 min. Each 4i cycle included the following steps: antibody elution (30 min in elution buffer), blocking (1 hr in 4i blocking buffer), primary antibody incubation (2 hr diluted in Intercept PBS Blocking Buffer (LI-COR)), and secondary antibody plus 6-Diamidino-2-phenylindole (DAPI, Sigma-Aldrich) incubation (30 min diluted in Intercept PBS Blocking Buffer), followed by multichannel imaging in imaging buffer. Cells were washed three times with PBS between steps. Elution buffer contained 0.5 M glycine, 3 M guanidinium hydrochloride, 3 M urea, and 70 mM TCEP-HCl, all from Sigma-Aldrich, diluted in MilliQ $H_2O$ and adjusted to pH 2.5. The 4i blocking buffer included 300 mM maleimide (Sigma-Aldrich) in Intercept PBS Blocking Buffer. Immediately prior to imaging, imaging buffer was prepared with 700 mM N-acetyl-cysteine (Sigma-Aldrich) and 100 mM HEPES, diluted in MilliQ $H_2O$ and adjusted to pH 7.4. pH adjustments were made with sodium hydroxide and hydrochloric acid. All steps were carried out at room temperature (RT) in the dark using an OT-2 liquid handler, with incubations performed on a see-saw rocker at 15 rpm. High-quality antibodies, previously validated for use in 4i[8], were chosen for labelling and are listed in Supplementary Table 2. Successful antibody elution was assessed at each cycle of the 4i process. This evaluation was performed by imaging elution control wells, which consisted of re-probing previously labelled, imaged, and eluted wells with secondary antibodies. Complete elution was verified for all cycles.

**EMT activation and Cyclic Immunofluorescence (CycIF).** For EMT activation, 3,000 A549 cells were seeded into 96-well glass-bottom plates. After 20 hrs, DMEM was removed, and cells were cultured with serum-free media for 24 hrs. Cells were then treated for 48 hrs with either 0.1 μg/mL human epidermal growth factor (hEGF, Cell Signalling Technology, #8916) or 0.01 μg/mL transforming growth factor beta 1 (TGF-β1, R&D Systems, #5036-WN) diluted in serum-free media.

Following activation, to perform standard 4-plex IF for the 4-plex ExIF EMT dataset, A549 cells were fixed with 4% PFA in tris-buffered saline (TBS) for 15 min and permeabilised with 0.1% Triton X-100 in TBS for 20 min. Cells were blocked for 1 hour with Intercept TBS Blocking Buffer (LI-COR) before being incubated with primary antibodies diluted in Intercept TBS Blocking Buffer for 2 hours. Cells were then incubated with secondary antibodies diluted in Intercept TBS Blocking Buffer for 30 minutes. Cells were washed three times with TBS between each step. All steps were carried out at RT using an OT-2 liquid handler. All antibody details are listed in Supplementary Table 2.

A modified version of the cyclic IF (CycIF) protocol was performed to generate the experimentally multiplexed EMT dataset[54]. After EMT induction, A549 cells were fixed with 4% PFA/PBS for 15 min and permeabilised with 0.1% Triton X-100/PBS for 20 min. Cells were then subjected to cycles of blocking with 4% BSA/PBS for 1 hour at RT, fluorophore-conjugated primary and DAPI labelling diluted in 4% BSA/PBS for 12 hours at 4 °C, multichannel imaging, and bleaching with 3% $H_2O_2$ and 20 mM NaOH in PBS for 1 hour at RT under a fluorescence lamp. Cells were washed three times with PBS between each step using a BioTek 50TS. This process was repeated over four cycles. Successful antibody bleaching was assessed at each cycle by imaging a control well that had been bleached but not re-labelled. Complete signal removal was verified for all cycles. Antibody details are listed in Supplementary Table 2.

**Confocal microscopy.** Images for standard IF and 4i datasets were acquired with a Nikon A1R confocal using JOBS automation for reproducible, unbiased multi-position, multi-well acquisition. Automated laser-based and image-based (using DAPI labelling) focusing were combined to ensure optimal, reproducible focusing. A 20x Plan Apochromat air objective (NA 0.75) was used to obtain images at 1024x1024-pixel resolution.

CycIF images were acquired with a Nikon AX-R2 confocal also using JOBS automation for standardised multi-well acquisition. As with the A1R, combined laser-based and image-based (DAPI) focusing was used to maintain focus. A 40x Plan Apochromat air objective (NA 0.75) was used to obtain images at 2048 x 2048-pixel resolution.

Pixel saturation was avoided through monitoring of high-low look up tables during laser power and gain setting to preclude pixel value clipping and resulting value ambiguity in the training of deep learning models. Raw 16-bit.nd2 image files were converted to TIFF format using custom Python scripts prior to quantitative analyses. Brightness and contrast were optimized with Fiji software (National Institutes of Health) for visualisation and presentation purposes only.

**Image analysis, virtual labelling, and downstream quantitative analyses**

**Pre-processing.** Background removal was performed on label-free DIC images by subtracting the median (40px kernel) filtered image in the experimentally multiplexed (A549 cell) dataset and subtracting a Gaussian blurred image ($\sigma = 0.5$) for the EMT (DU145 cell) dataset due to the different imaging artefacts in each dataset. Contrast enhancement was then applied to DIC images, allowing 35% of the pixels to become saturated. Images were then converted to 8-bit/pixel by performing linear scaling, projecting the min-max to 0–255, respectively in Fiji software. Fluorescence images were converted to 8-bit by performing linear scaling with respect to the whole dataset. The top and bottom 10% of pixels across all fluorescence images were treated as outliers and the min-max of remaining pixels in the dataset were projected to 0–255 respectively (saturating the outliers). All images were then resized by half using bilinear interpolation, before being cropped into patches measuring 256 × 256. In the experimentally multiplexed EMT dataset, these images were additionally denoised using BM3D[57] (psd=15).

This resulted in a total of 583 patches of DU145 cells, each with – DIC, α-Tubulin, DNA, CoxIV, Fibrillarin, GM130, F-Actin, β-Catenin, and NF-κB channels; 296 patches of A549 cells per well in the 4-plex ExIF EMT dataset, each with DIC, DNA, F-Actin, and β-catenin as well as one of the following variable markers: E-Cadherin, EpCAM, N-Cadherin, PTEN, Vimentin, CD44total, CD44std, CD44v9; and 576 patches per well of A549 cells in the experimentally multiplexed EMT dataset, each with – DIC, DNA, Actin, β-catenin, β-tubulin, CD44, E-Cadherin, EpCAM, N-Cadherin, pan-cytokeratin, pSMAD23, Vim, Zeb1, ZO1. To simulate the structure of ExIF, in addition to DIC, DNA, F-Actin, and β-catenin, a single 'variable' marker from one of - β-tubulin, CD44, E-Cadherin, EpCAM, N-Cadherin, pan-cytokeratin, pSMAD23, Vim, Zeb1, ZO1 was also 'assigned'. This results in 4 'assigned' markers per well with each panel. In both EMT datasets, each panel was labelled once in control, EGF, and TGF- β1-treated wells. All 'ExIF' training done in the experimentally multiplexed EMT dataset follows this simulation using only the markers available/assigned. Details summarised in Supplementary Table 3.

**Virtual labelling model design and training.** Three models were considered for virtual labelling. First, a model based on U-Net[31] was used, consisting of down-convolution blocks with skip connections to up-convolution blocks. Convolutions were used with a 4×4 kernel, 2×2 stride and 1×1 padding, followed by ReLU activation (leaky for down-convolutions with 0.2 negative slope) and then batch normalisation ($\varepsilon = 10^{-5}$, momentum = 0.1) apart from the first and last convolutional layers. Layer size was initialised at 64 filters and doubled until 512 filters. A total of 8 down-convolution layers were used with corresponding 8 up-convolution layers; deeper than the original U-Net as we informally found small improvements with increasing depth. 0.5 Dropout regularisation was added for 3 of the layers before the deepest layer. Tanh activation was used after the final up-convolution. L1 loss was used as the objective function.

Second, the cGAN model[32] utilised the U-Net backbone described as a generator network and was extended with adversarial training. The discriminator network was comprised of down-convolutions with a 4×4 kernel, 2×2 stride and 1×1 padding starting with 64 filters and doubling until 512 filters. Each convolution layer was followed by leaky ReLU activation (0.2 negative slope), and batch normalisation ($\varepsilon = 10^{-5}$, momentum = 0.1) apart from the first layer. The final down-convolution layer had 1×1 stride. A final classification layer was also added with 1×1 stride. We combined L1 loss with LSGAN loss which is the mean squared error of the discriminator's prediction.

Both U-Net and GAN models were trained for 50 training epochs with a learning rate of 0.0002 before linearly decaying for another 50 epochs as conducted by others[30].

Finally, a residual vision transformer model – Residual vision transformer (ResViT) was used[30]. It consists of an encoder and decoder module consisting of CNN blocks, and a bottleneck in between composed of a series of aggregated residual transformer (ART) blocks composed of multi-head self-attention and convolutions. The network (without the transformer) pretrained on the public ImageNet21k dataset and was further trained for 50 epochs with a learning rate of 0.0002 before linearly decaying for another 50 epochs. The whole network (including the transformer) was then trained for another 25 epochs with a learning rate of 0.0002 before linearly decaying for another 25 epochs[30].

Optimisation was performed using ADAM ($\beta_1 = 0.5$, $\beta_2 = 0.999$) for all three models[58].

Model training was conducted using 5-fold cross-validation, stratified with respect to treatment conditions for EMT-activated cells. Data augmentation by vertical and/or horizontal flipping expanded the number of unique images by a factor of four. To normalise the data, image pixel intensities were scaled between [-0.5, 0.5]. Model training was conducted using an NVIDIA V100 GPU taking roughly 4 hours per

fold for label-free ExIF and roughly 5 hours for full-ExIF resulting in about 20-25 hours for all 5 folds for our dataset. As each additional variable marker requires an additional model, sequential training would scale linearly with the number of variable markers.

**Segmentation and Feature extraction.** For downstream analysis and performance evaluation, features were extracted per cell using a custom workflow in CellProfiler[59]. First, cell segmentation was performed using the default 'Cells' Cellpose model on experimental DNA (DAPI) and F-Actin (Phalloidin) channels. Objects in contact with the image border were excluded to remove partial cells. In total, per cell, we extracted pairwise marker signal colocalization (Pearson's correlation coefficient based on per pixel intensities) plus 15 intensity, 75 radial distribution, and 52 texture features (see Supplementary Table 1). For the 8-plex experimentally multiplexed DU145 dataset (section "Introduction"), 45,780 cells were assessed from a single biological replicate with 8 technical replicates (wells). For the 4-plex ExIF EMT dataset, 33,572 cells were analysed from a single biological replicate across 24 distinct treatment and labelling conditions (wells). For the experimentally multiplexed EMT dataset, 137,893 cells were analysed from a single biological replicate across 30 distinct treatment and labelling conditions.

**Evaluation.** Predicted labels were evaluated at the image-level using the structural similarity index measure[34] (SSIM) for a 7×7 window and Pearson's correlation coefficient between corresponding pixels (Pixel-PCC). Co-localisation of predicted and experimental labels was calculated using Pearson's correlation coefficient of pixels within individual segmented cells across all cells in CellProfiler. Similarly, Pearson's correlation coefficient was used to measure the correlation between features extracted from experimentally and virtually labelled cells across all cells (Feature-PCC).

**Classification.** Cells segmented from the EM dataset (section) were categorised into three classes corresponding to treatment conditions – control, EGF (48hrs), TGF- β1 (48hrs). For a given population, cells were then split into training and test sets using stratified 5-fold cross-validation, maintaining class proportions. Standard scaling was applied to each feature, using the training set to calculate mean and standard deviation parameters. In the 4-plex ExIF EMT dataset, principal component analysis (PCA) was fit on the training set and performed across the dataset using Minka's maximum likelihood estimation[47,48] (MLE) to select a number of principal components. Features were then used to classify treatment condition using a support vector machine (SVM), including a radial basis function kernel or linear kernel for 4-plex and experimentally multiplexed EMT datasets, respectively. Prediction performance was evaluated using the mean F1-score across the folds.

Populations were defined by the variable marker used for labelling. For each population, we classified using morphological and neighbourhood features, as well as features extracted from (1) only experimentally labelled common markers, (2) experimentally labelled common markers and the variable marker, (3) common markers and a variable marker predicted using ResViT (the top performing model) with DIC input, and (4) experimentally labelled common markers and all variable markers predicted using ResViT, using label-free and common markers input, and where available (5) all common and variable markers experimentally labelled.

**Embedding and pseudotime analysis.** Unsupervised embeddings were created for the cells in the EM dataset using intensity, radial distribution, and texture features extracted from (1) three common markers; (2) All markers predicted using DIC only; (3) The three common markers experimentally determined, and all variable markers

predicted using DIC and the three common markers as input; and where available (4) all common and variable markers experimentally labelled. In the 4-plex ExIF EMT dataset, features were standard scaled and PCA transformed, using Minka's maximum likelihood estimation (MLE) to select an optimal number of principal components. Transformed features were then embedded using T-SNE[47], UMAP[48], or PHATE[46] using default parameters. Extreme outlier cells were removed solely for compact graph visualisation in the final figures. Pseudotime analysis was conducted on complete data using scFates[52] employing a 25-node curve with root determination based on the control treatment condition of the cells. In the experimentally multiplexed EMT dataset, standard scaled features were then embedded using T-SNE[47], UMAP[48] using default parameters or PHATE[46] with knn=20. Extreme outlier cells were removed solely for compact graph visualisation in final figures. Pseudotime analysis was conducted on complete data using scFates[52] employing a 8-node curve with root determination based on the control treatment condition of the cells.

### Statistics & Reproducibility
Experiments subject to statistical testing were repeated using 5-fold cross validation with significance tests conducted using Welsh's two-sided t-test (unequal variance assumptions) as noted in relevant figure legends. $P$ value $< 0.05$ was considered as statistically significant. No data was excluded.

### Reporting summary
Further information on research design is available in the Nature Portfolio Reporting Summary linked to this article.

### Data availability
The raw and pre-processed (as described in methods) images, performance metrics, and Cellprofiler outputs generated in this study have been deposited in a Figshare repository[60] (https://doi.org/10.6084/m9.figshare.26500210).

### Code availability
Code used to train and apply models is made available at https://github.com/CancerSystemsMicroscopyLab/VirtualLabelling.

The version used in this study has been published on Zenodo[61] (https://doi.org/10.5281/zenodo.15172098). We have also provided sample images and an IPython notebook example to use virtual labelling with other datasets. The same example is provided as a Google Colab notebook which only requires the upload of additional datasets and removes any necessary hardware/dependency set up. Instructions for the use of both notebooks are included in the Github repository. Code for image preprocessing have also been included. An ImageJ macro is provided to perform background subtraction for DIC images as described in methods. A Python script is provided for the conversion of 16-bit to 8-bit images with contrast enhancement across the whole dataset (per marker as opposed to per image). Code used to reproduce analysis presented in the manuscript is also included.

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

## Acknowledgements

I.G. and F.V.K. are supported by Australian Government Research Training Program (RTP) Scholarships. M.D. is supported by the UNSW University Postgraduate Award. F.V.K. receives a Top-Up award from SPHERE Cancer CAG and Cancer Institute NSW. Y.Z. receives a UNSW TF scholarship. J.G.L. is supported by a University of New South Wales Scientia Research Fellowship, a Ramaciotti Biomedical Research Award, an ARC Development Project grant (DP170103599), NHMRC Ideas Grants (GNT1184009, GNT2012848, GNT2028506), and a Tour de Cure Pioneering Grant (RSP-547-FY2023). This research was undertaken with the assistance of resources and services from the National Computational Infrastructure (NCI), which is supported by the Australian Government.

## Author contributions

I.G., E.M., and J.G.L. conceived the paper and project design. Virtual labelling and all downstream analyses conducted by I.G., with input from D.P.N. Preparation, multiplexed labelling and imaging of DU145 cells conducted by F.V.K. and Y.Z. Preparation, labelling, and imaging of multiplexed and 4-plexed A549 cells conducted by M.D. and K.N., respectively. Design and writing of the manuscript was led by I.G. and J.G.L., with contributions from F.V.K. and D.P.N. Significant conceptual development and editing by E.M. and F.V.

## Competing interests

The authors declare no competing interests.
