## [Transparent Peer Review file · Nature Communications]

Extensible Immunofluorescence (ExIF) accessibly generates high-plexity datasets by integrating standard 4-plex imaging data

Corresponding Author: Dr John Lock

Version 0:

Reviewer comments:

Reviewer #1

(Remarks to the Author)

Summary of the Work

The authors present a framework, termed Extensible Immunofluorescence (ExIF), which utilizes standard 4-plex panels for multichannel immunofluorescence imaging to generate a unified dataset. This dataset includes multiplexed virtual markers generated through deep learning (DL) techniques. The framework is designed to provide multichannel data that facilitates the study of complex cellular processes at the molecular level.

The work compares three deep learning-based generative models, with a particular focus on ResViT, a transformer-based architecture. ResViT demonstrates superior performance compared to UNet and cGAN, as measured by the Pearson's Correlation Coefficient (PCC) between reference images and those generated by the models at the pixel level.

The applicability of the dataset generated by ExIF is demonstrated through an experiment on the epithelial-mesenchymal transition (EMT), showing its use in phenotype classification, acquisition of heterogeneous phenotype manifolds, and pseudo-temporal inference of molecular marker dynamics.

=====

General impressions

The authors have undertaken a thorough and commendable effort in presenting the key aspects of their work. The tools used for data analysis and the conclusions drawn are appropriate. The manuscript is well-written, and the motivation and objectives are clearly stated. However, while the sections are logically structured, it is not immediately clear how the computational methods described contribute to the construction of the integrated dataset using ExIF.

The experiments described in the results section are detailed, but it is difficult to trace how some the computational elements are applied within ExIF. The manuscript would benefit from clearly and concisely answering the following question:

How are the computational methods described in the results section used in the different stages of applying ExIF in a cellular biology study, from virtual labelling to consequent application?

This question should at least be addressed in the context of the ResViT model application.

The manuscript demonstrates that several state-of-the-art techniques have been applied to specific experimental problems, including:

- Virtual labeling and generation of virtual EM markers using ResViT.
- Classification of cell treatments.
- Mapping of heterogeneity and dynamics of markers obtained via virtual labeling.

In all experiments, virtual labeling is the common thread, and its effectiveness is convincingly demonstrated. However, it is debatable whether the combination of computational methods used can be presented as a part of the framework for data

integration. The computational tools applied are not exclusive to ExIF and could be applied to any multichannel dataset obtained through other methods. It would be clearer if ExIF were defined as a process for generating a multichannel dataset for cellular studies rather than as a framework for data integration.

Nevertheless, the authors should be recognized for successfully presenting a practical workflow for multichannel dataset analysis in the context of cellular studies. This review aims to assist in the manuscript's revision and improve the quality of its presentation while acknowledging the substantial work done by the authors.

Consequently, I recommend conditional acceptance of the manuscript for publication, provided the following points are addressed, and the manuscript is deeply revised accordingly.

Next, comments, appraisals, and suggestions are addressed to the authors with the aim of guiding the manuscript's revision and contributing to its improvement, while acknowledging the remarkable work already done. To ensure clarity, the comments and suggestions follow the manuscript's section order, indicating the specific lines and graphic elements for revision. The comments are categorized based on their importance.

=====
Main issues

[Introduction]

1. [lines 62-64]: The manuscript should clarify whether the capability to include an unlimited number of molecular markers is an inherent property of ExIF or if it is due to the structure of the individual marker generation models. Is this property derived from the way the models are trained to incorporate additional input channels?
2. Figure 1-C: It is unclear whether the represented model is a single model generating all virtual markers (VC1-8) or separate models for each marker. Clarifying this would improve the understanding of its application.
3. [lin. 111-115]: Although ExIF is described as a shift from label-replacement to dataset integration, the work presented seems more aligned with a multi-label-replacement paradigm. The authors should include a brief description of the "simple experiment" mentioned in these lines.
4. [lin. 116-118]: The claim that the proposed virtual labeling model surpasses the state-of-the-art should be reviewed and supported with stronger evidence. The statement could be questioned if alternative metrics (e.g., SSIM) are considered. Although three architectures are compared, only one metric (i.e., PCC) is presented to verify the best model performance. Additionally, performance comparisons with previous work about virtual labelling are missing.
5. [lin. 122-125] It is not surprising that models with more input information (channels) perform better, as seen in 4-plex IF studies. This result should not be presented as specific to ExIF. The manuscript should compare ExIF's advantages quantitatively with other methods for obtaining multichannel (multi-marker) data.

[Results]

6. [lines 136-137]: The claim that ResViT is the first transformer-based architecture for virtual labeling should be reviewed considering recent studies (e.g., DOI: 10.1109/TMI.2020.2968504 and DOI: 10.1117/1.JBO.29.3.036004).
7. [lines 137-143]: Figure 2A-left-B-C-D-E should be presented separately as it pertains to the development and testing of computational elements but does not constitute a functional phase of ExIF itself. Combining these elements in one figure makes it harder to follow the explanation. Additionally, it would help to clarify the connection between this figure and the process shown in Figure 1.
8. [lin. 157-159]: While experiments using only the DIC channel as input show an advantage for ResViT, this does not guarantee that the differences between architectures will remain significant when more input channels are used. With more input data, all architectures should perform better, potentially reducing the observed differences and even their statistical significance.
9. [lines 166-172]: In the virtual labeling experiment, input from a DIC channel and seven markers is used to predict an eighth marker, but it is not clear which marker is inferred. Clarify this.
10. [lin. 236-238] Clarify whether ExIF generates a single model for virtual labeling of all EM state markers or multiple models, one for each marker. Indicate whether pre-trained models were fine-tuned with data from each experiment. This explanation could be detailed in the methods section.
11. [lin. 262-264] It should be clarified to what extent the superior performance obtained is attributed to the use of the ExIF dataset, rather than being an emergent property of the capacity of generative virtual labelling models (regardless of the method followed to obtain the dataset).
12. [lin. 520-522] Review the statements made so that they are justified with arguments based on quantitative comparisons with the alternatives mentioned in the text (i.e., related works).

[Discussion]

13. [lin. 585-586] Include a concise list of the innovations claimed by the authors and explain why they are considered novel in the context of virtual labeling.

[Methods]

14. [lin. 634-645] Ensure that detailed information is provided regarding the datasets used in the experiments with A549 and DU145 cells. It would be helpful to present this information in a tabular format.

15. [lin. 672-673] It is especially important to highlight the computational cost required for the application of ExIF since this value allows a fair comparison with other alternative methods of obtaining multimodal datasets. That is:

- The total time spent in a study carried out (described in the sections contained in line 227-522) should be included following the phases proposed in the workflow using ExIF and this total time should be broken down into its phases. In the training phase of the models, also show the total time spent ¿20 hrs. = 4 hrs./fold x 5 fold?

- Indications on the dependence of the total processing time on the total number of channels (markers) used in ExIF.

- Differentiate between the times required when using label-free ExIF and full ExIF application models.

- It might even be highly informative to include the total cost of the hardware equipment used on which the models are trained.

=====
Secondary issues

1. Review all figures to improve readability (e.g., size), particularly for boxplots, which may benefit from being replaced with tables showing central tendency, dispersion, and t-test values.

2. Figure titles are often overly detailed and redundant. The titles should not repeat information already explained in the text or within the figure itself.

3. [Fig. 1-D] The term "Model Training" is used; should "inference" also be mentioned?

4. [line 151] The phrase "...Person's correlation..." should be corrected to "...Person's correlation coefficient...".

5. [Fig. 4-C] Recheck the confusion matrices for rounding errors, particularly in the lower rows for Full ExIF (center) and Difference (right).

6. [lin. 647] Replace "...virtual training..." with "...virtual labeling...".

7. [lin. 690] The phrase "...no variance assumptions..." should be corrected to "...unequal variance assumption...".

(Remarks on code availability)

The code hosted in the GitHub repository is incomplete and only covers simple image preprocessing and an example of virtual labeling with pre-trained ResViT models. The code does not include the analyses presented in the manuscript. As a conclusion, it is not possible to reproduce the results of the study based on the code provided. However, the confidentially provided datasets do fit the content described in the manuscript.

Reviewer #2

(Remarks to the Author)

The authors present a new method termed "Extensible Immunofluorescence" (ExIF), which utilizes 4-plex immunofluorescence and DIC images to predict additional markers, thereby generating a highly multiplexed dataset. The standard setup described includes three constant markers and one of eight variable markers. The constant markers are used to train a deep learning network to predict the image of the fourth, variable marker. The authors compare the performance of different network architectures (U-Net, CGAN, and ResViT), identifying ResViT as the most suitable for this task. Through a set of control measurements, they assess the accuracy of this approach in predicting features commonly used to characterize single cells in immunofluorescence experiments (Cellprofiler). Furthermore, they apply this method to create a virtual multiplexed dataset of control cells and cells undergoing epithelial-mesenchymal transition (EMT) with EGF or TGF- β 1 stimulation. As a final validation, they test their virtual multiplexed dataset in a classification challenge (predicting the cellular condition) and use manifold extraction (PHATE) to illustrate the trajectory of cell states along the EMT axis.

In summary, the authors present a new interesting experimental and computational approach to the multiplexed fluorescence experiment. Although, I would personally question the strong assumption of the authors that multiplexed imaging is complex and inaccessible, I do believe that there is space for additional approaches. The approach presented in this manuscript is a form of clever 'feature engineering'. When the model predicts the virtual channel based on the anchor channels, it is essentially finding a way to represent the information that was already present in those anchor channels in a new form. This process doesn't add new information; rather, it redistributes or reinterprets the existing information into a form that resembles the virtual channel. This is a form of decomposition and can be very beneficial in the downstream analysis.

My primary criticism concerns the tone of the paper. The manuscript is written from the perspective that this approach can fully replace a multiplexed experiment, aiming to prove its feasibility. In reality, this method will always offer an intermediate level of information in describing a set of cells in a multidimensional space of n markers. It occupies a middle ground between an experiment with a limited number of measured markers and a fully multiplexed experiment. The value of this approach lies in its ability to estimate most markers for most cells using the ExIF method, albeit this comes at the cost of a higher error rate compared to a real measurement in a standard multiplexed experiment. I strongly encourage the authors to include this notion in the manuscript leaving a potential user to evaluate if this new approach is suitable for their needs.

In the trajectory inference experiment presented in Figure 5, the authors compare manifolds derived from a standard fluorescence experiment (Standard Manifold) with those from Label-free ExIF and Full ExIF approaches. It is unsurprising that manifolds incorporating more information from the measurement phase perform better. However, a more informative comparison would be between the Full ExIF Manifold and a manifold obtained from a fully multiplexed experiment. Although the authors emphasize the potential to study cellular heterogeneity with the Full ExIF Manifold, it ultimately reveals only a single trajectory, indicating limited complexity. It is plausible that the same conclusion could be reached using three wells (one per condition) in a fully multiplexed experiment.

To elaborate, the experiment currently measures DAPI, two anchor markers, and eight variable markers. In a fully multiplexed experiment, this could be replicated with four rounds of immunofluorescence, using fewer antibodies for anchor markers and the same number of antibodies for variable markers, while requiring fewer cells and less reagent. This would result in a cheaper experimental setup. However, ExIF has the advantage of being completed in a single round, reducing the experimental time by a factor of four. Comparing these two approaches directly is more appropriate than comparing ExIF to a 4-plex fluorescence experiment, and the advantages and disadvantages of the new method should be clearly outlined.

It could be argued that ExIF's ability to analyze a significantly larger number of cells may reveal more complex alternative trajectories or distinct subpopulations. However, it seems unlikely that if such alternative states were not detectable in a single well, the networks generating the missing signals would accurately predict the missing markers to the extent that they would be distinguishable in the final manifold. I would welcome it if the authors could provide evidence to refute this notion.

In the clustering analysis presented in Figure 4, the authors use the ExIF dataset to extract cell features, applying PCA and SVM to cluster the data and evaluate the accuracy of cell state prediction. They claim that this experiment "reveals a remarkable increase in classification accuracy due to virtual labelling-enabled data integration, despite both analyses ultimately relying on identical underlying image data (the three general markers)." However, this statement is not entirely accurate. First, the ExIF analysis includes additional measurements, as the real measurements of the fourth channels were not excluded, based on my understanding. Second, the experiment is designed such that cells within a single image share the same experimental condition. This setup could allow the network to detect correlations in signal levels between cells within the same image, meaning the virtual channel could encode this information. In contrast, cell classification is performed on a feature set that lacks any information about the cell neighborhood. Consequently, the classification results are confounded by the introduction of cell neighborhood information into the analysis. Therefore, I do not believe this experiment effectively demonstrates the merits of the ExIF approach.

(Remarks on code availability)

Version 1:

Reviewer comments:

Reviewer #1

(Remarks to the Author)

After a careful reading of the updated version of the manuscript, I find no need for a further revision of the content. In this updated version the authors have successfully enhanced the clarity and depth of their work, addressing all the concerns and suggestions raised by the peer reviewers. The new manuscript version reflects several improvements:

+ Stronger scientific foundation: The introduction and discussion sections now provide a more comprehensive context, helping readers understand how ExIF fits into the broader field of cellular analysis and fluorescence imaging.

+ Improved structured methodology: The revised version offers a clearer step-by-step breakdown of the ExIF framework, making it easier to understand the approach.

+ More rigorous validation: The updated version includes a more detailed performance analysis, comparing ExIF to both computational and experimental multiplexing methods.

+ Enhanced figures and explanations.

+ Stronger discussion of limitations and future directions: The authors acknowledge the current challenges of ExIF and

suggest ways to improve it.

+ More complete description of software and datasets. The information provided in the repositories ensures the completeness and reproducibility of the results.

Therefore, the updated manuscript is a significantly improved version of the work offering a more robust, transparent, and informative presentation of ExIF.

(Remarks on code availability)

Reviewer #2

(Remarks to the Author)

The authors have thoroughly addressed my questions and incorporated the suggested revisions. I believe the current version clearly presents ExIF as an experimental approach that bridges 4-plexed and fully multiplexed imaging strategies. The manuscript is now well-positioned to be a valuable resource for the community.

(Remarks on code availability)

REVIEWER COMMENTS

=====
Reviewer #1 (Remarks to the Author):

=====
General impressions

The authors have undertaken a thorough and commendable effort in presenting the key aspects of their work. The tools used for data analysis and the conclusions drawn are appropriate. The manuscript is well-written, and the motivation and objectives are clearly stated. However, while the sections are logically structured, it is not immediately clear how the computational methods described contribute to the construction of the integrated dataset using ExIF.

The experiments described in the results section are detailed, but it is difficult to trace how some the computational elements are applied within ExIF. The manuscript would benefit from clearly and concisely answering the following question:

How are the computational methods described in the results section used in the different stages of applying ExIF in a cellular biology study, from virtual labelling to consequent application?

This question should at least be addressed in the context of the ResViT model application.

The manuscript demonstrates that several state-of-the-art techniques have been applied to specific experimental problems, including:

- Virtual labeling and generation of virtual EM markers using ResViT.
- Classification of cell treatments.
- Mapping of heterogeneity and dynamics of markers obtained via virtual labeling.

In all experiments, virtual labeling is the common thread, and its effectiveness is convincingly demonstrated. However, it is debatable whether the combination of computational methods used can be presented as a part of the framework for data integration. The computational tools applied are not exclusive to ExIF and could be applied to any multichannel dataset obtained through other methods. It would be clearer if ExIF were defined as a process for generating a multichannel dataset for cellular studies rather than as a framework for data integration.

Response to Reviewer 1; Point 0 (RR1;P0)

We thank the Reviewer for this thoughtful critical feedback.

It is indeed important to distinguish between the ExIF strategy presented herein (as schematised in Figure 1), and the statistical / machine learning methods (classification, manifold learning of cell heterogeneity) we have used to test / demonstrate the efficacy of the ExIF strategy. These latter are not depicted in the ExIF overview figure (Figure 1), as they are not part of the proposed ExIF framework, but are used to test it and display how it can be used.

The ExIF strategy itself comprises an experimental component – extensible immunofluorescence (Figure 1A) which produces 4-plex panels (Figure 1B), and a computational component – the training of virtual labelling models with multi-channel inputs (Figure 1C) producing an integrated dataset (Figure 1D). The novel (though simple for users to implement) extensible labelling concept of designing 4-plex labelling panels containing explicit ‘anchoring’ and ‘variable’ marker channels enhances the subsequent use of deep learning models for virtual labelling to generate integrated datasets. By first establishing the enhanced performance achieved by using multiple (fluorescence and label-free) channels as inputs for virtual labelling models (Figure 2) and choosing a suitable deep learning model (ResViT), we exemplify this combined experimental and computational ExIF strategy using EMT-inducing perturbations as a ‘known’ biological test-bed (Figures 3-5).

To quantify the efficacy of ExIF, we employ gold-standard methods of single cell image quantification (CellProfiler) followed by gold-standard multivariate analyses (used in omics and image cytometry) for classification (Figure 4) and manifold learning of cell heterogeneity (Figure 5). These downstream analysis steps are not presented as components of ExIF (i.e. they are not schematised in the overview of ExIF workflow, Figure 1), but rather as means to quantitatively assess the utility of ExIF in an example biological use-case – as the Reviewer suggests. We nonetheless have now provided the code associated with these ‘downstream analyses’ so that our results can be replicated, and so that users can immediately employ these analytical tools (which are indeed not unique for ExIF) for analyses of their own ExIF datasets.

We appreciate the reviewer’s note that our approach is both a method for data integration and for generating a multichannel dataset; we have indeed deeply considered and also tested which of these conceptual forms is best for communicating the ExIF strategy. One important difference between our approach and the way that multi-channel label replacement / generation has been performed previously is that we are not training virtual labelling models (with an initial *training dataset*) that are then intended to be used for prediction in subsequent *application datasets* (the approach that we term ‘replacement’). Although our approach could be used to this end, we here focus on the capacity to integrate a multi-panel 4-plex dataset to make it into a multiplexed dataset, with training

and the application of integration achieved *within a single dataset*. This is a different concept, which enables final integrated plexity to be greater than that of the input datasets (in fact, plexity is technically unlimited in this approach unlike other referenced methods that use multiplexed inputs to predict the same plexity outputs). We feel this encourages a different kind of use for virtual labelling that provides new benefit to a broader community of users, especially those that are limited to standard 4-plex imaging but nonetheless wish to assess a large number (theoretically unlimited) of markers. This ability to multiplex based only on 4-plex imaging data differs in practical terms from previously described virtual multiplexing methods, since all such approaches that we are aware of are presented in use cases where real multiplexed datasets are the training set for subsequent multiplexed label generation. This requires that users establish the capacity and protocols for multiplexed labelling, which is a significant barrier to entry for many users (hence the usefulness of virtual multiplexing). We have added text (line 64) that clarifies this point of difference to prior multiplexed generation approaches, referencing other multiplexed virtual labelling methods as also suggested, as follows:

“In the context of tissue imaging and digital pathology, efforts have gone further, using deep learning approaches to virtually reconstruct multiplexed data from a smaller number of input channels²⁴⁻²⁶ or even from hematoxylin and eosin (H&E) staining²⁷. Traditional machine learning methods have also enabled imputation of quantified multiplexed marker features (e.g. expression level per cell), though this approach fails to recapitulate functionally vital information on subcellular marker localisation²⁸. These methods have been presented for recreation of virtual data matching the marker plexity of the training dataset, meaning that high plexity virtual labelling first requires generation of experimentally multiplexed labelling data. Such approaches thereby implicitly sustain dependence on multiplexed labelling methods that are, as yet, not employed by the majority of fluorescence imaging practitioners.

In contrast, here we have developed ‘Extensible Immunofluorescence’ (ExIF), a framework that can integrate a theoretically unlimited number of markers from multiple panels of standard (4-plex) IF labelling, thereby increasing biological insights accessible from downstream analyses without requiring experimental multiplexing for prior training of virtual labelling models.”

With the above considerations, and based on presentation and discussion of our ExIF concept with both imaging and multi-omics practitioners, we believe that emphasising the integrative nature of our approach has the broadest interpretability but also, perhaps more importantly, highlights conceptual and methodological commonalities (such as the generation of explicit ‘anchoring’ features) not previously emphasised between the strategies emerging for multi-omic data integration and the strategies used for computational image multiplexing. We seek to strengthen recognition of these commonalities by employing the language of data integration (i.e. mosaic data integration;

line 96) that has been defined in seminal multi-omics integration studies / perspectives. We believe that this highlights the potential to develop integrative strategies for the imaging field that may evolve beyond what is presented herein, based on the conceptual formalisms emerging in multi-omics (e.g. Argelaguet et al doi.org/10.1038/s41587-021-00895-7). Hence, whilst we have indeed tried communicating this approach under both of the conceptual guises identified by this reviewer (multiplexed generation vs multiplexed integration), we are convinced that the integrative conceptualisation communicates with a broader audience and provides the greatest scope for constructive growth around these concepts.

Nevertheless, the authors should be recognized for successfully presenting a practical workflow for multichannel dataset analysis in the context of cellular studies. This review aims to assist in the manuscript's revision and improve the quality of its presentation while acknowledging the substantial work done by the authors.

Consequently, I recommend conditional acceptance of the manuscript for publication, provided the following points are addressed, and the manuscript is deeply revised accordingly.

Next, comments, appraisals, and suggestions are addressed to the authors with the aim of guiding the manuscript's revision and contributing to its improvement, while acknowledging the remarkable work already done. To ensure clarity, the comments and suggestions follow the manuscript's section order, indicating the specific lines and graphic elements for revision. The comments are categorized based on their importance.

=====

Main issues

[Introduction]

1. [lines 62-64]: The manuscript should clarify whether the capability to include an unlimited number of molecular markers is an inherent property of ExIF or if it is due to the structure of the individual marker generation models. Is this property derived from the way the models are trained to incorporate additional input channels

RR1;P1

The extensible labelling panel design proposed in the ExIF framework (schematised in Figure 1A) enables structured virtual labelling such that the number of channels integrated is only limited by the number of variable channels in a single dataset, which can, in the presented schema, be increased by adding additional wells of 4-plex labelling. Thus, using the extensible labelling design for actual experimental labelling in

combination with the virtual labelling approach using multiple ‘anchoring channel’ inputs plus additional ‘variable channels’ as targets, high-quality virtual labelling can then be used to integrate a theoretically unlimited (as many wells as the user wishes) number of markers.

We clarify this in text (line 88), adding *“This ‘extensible labelling’ allows a theoretically unlimited number (plexity) of molecular markers to be added in parallel for subsequent integration”*, making it clear that this is a property of ExIF panel design that is leveraged through the virtual labelling approach.

2. Figure 1-C: It is unclear whether the represented model is a single model generating all virtual markers (VC1-8) or separate models for each marker. Clarifying this would improve the understanding of its application.

RR1:P2

Thank you for the input, we have added multiple icons for the model colour coded according to the outputs, making it visually clearer that each output corresponds to a separate model. Further, the legend text for Figure 1C states that *“A separate deep learning model is trained for each variable channel using real labelling as training targets from each respective well and anchoring channels as input. Trained models are then used to generate predictive ‘virtual channels’ for each variable channel in all cells in all wells”*. We have also modified the main text (line 90) to state *“the anchoring channels are then used as consistent inputs for multiple generative DL models that are each trained to produce ‘virtual labelling’^{11,12} of a single variable channel (Figure 1C).”*

3. [lin. 111-115]: Although ExIF is described as a shift from label-replacement to dataset integration, the work presented seems more aligned with a multi-label-replacement paradigm. The authors should include a brief description of the “simple experiment” mentioned in these lines.

RRP:P3

Thank you for the feedback. Regarding the key differences between label replacement and dataset integration, we have detailed our rationale and reasoning for using the ‘integration’ conceptual framework in **RRP:P0**.

Regarding the “simple experiment” – we believe the reviewer may be referring to the term “single experiment” (previously line 114 now 85), this refers to the capacity of ExIF to achieve model training and dataset integration based on a single experiment, as schematised in Figure 1, a capability that differs (as noted above) from the need for experimentally multiplexed training datasets preceding the application of virtual labelling

(in previous multiplexed label replacement applications, see **RR1:P1**). We seek to provide further clarity on this point as follows (line 97): *“As no independent training set is required, ExIF integration and subsequent interrogation can happen all within the same experiment.”*

In case the reviewer was referring to the ‘simple implementation’ (previously line 117 now 133), which refers to the ‘EM state’ experiment (Figures 3-5), whose details are provided (line 207): *“To assess the utility of ExIF for interrogating EM cell state plasticity, we amplified EM state diversity in A549 lung cancer cells by comparing control cells to those treated for 48 h with either EGF (0.1µg/mL) or TGF-β1 (0.01µg/mL); growth factors known to weakly or strongly (respectively) drive epithelial-to-mesenchymal transition (EMT) in A549 cells³⁵. To readout EM state per cell, we labelled and imaged eight different 4-plex IF panels across each treatment condition, following the extensible labelling schema schematised in **Figure 1A&B**. Specifically, each panel consists of recurring ‘general’ markers (DNA, F-Actin and β-Catenin), and a single variable EM state marker (E-Cadherin or EpCAM or N-Cadherin or PTEN or Vimentin or CD44total or CD44std or CD44v9⁴⁰⁻⁴⁵ (**Figure 3A**).”*

As noted, this experimental design is also depicted schematically in **Figure 3A**.

4. [lin. 116-118]: The claim that the proposed virtual labeling model surpasses the state-of-the-art should be reviewed and supported with stronger evidence. The statement could be questioned if alternative metrics (e.g., SSIM) are considered. Although three architectures are compared, only one metric (i.e., PCC) is presented to verify the best model performance. Additionally, performance comparisons with previous work about virtual labelling are missing.

RR1:P4

We appreciate the feedback and have reviewed our statements in this section.

Our purpose by presenting the ResViT, U-Net and cGAN architectures is now more clearly presented as showing that the ExIF framework can be generalised for use with different DL architectures suitable for image generation, being fundamentally agnostic to the specific model used. This point is made to highlight that the ExIF framework described here can take advantage of future improvements in the performance of (new) DL model architectures. We now only present the ResViT data in main Figure 2, with U-Net and cGAN data in Supplementary Figure 1 – removing any emphasis on inter-model performance comparison, and hence any notion of DL model performance benchmarking. In addition to these changes in data presentation, we have included the following revised text (line 156): *“We here assess ExIF using a residual vision transformer (ResViT), making use of state-of-the-art self-attention³⁰. The ExIF strategy is nonetheless generalisable for use with any DL model suitable for effective virtual labelling, meaning that the same framework can leverage future advances in DL model architecture that*

*further improve technical virtual labelling performance. To demonstrate the DL model-agnostic nature of the ExIF framework, we show similar quantitative results in terms of virtual marker fidelity, as well as model performance increases with additional input channels, for two other DL architectures that are commonly used in virtual labelling (U-Net³¹, cGAN³²; **Supplementary Figure 1A**).”*

Reflective of these changes, it is not our intention to benchmark various architectures (in part because these are in extremely rapid flux), hence the revised messaging in this section to focus only on generalisability, not on relative performance. Additionally, we note that any such benchmarking would be uninterpretable (due to differences between underlying datasets if the raw performance numbers from different studies were directly contrasted). However, our inclusion of results from standard representative architectures with the different input strategies developed in this study establishes the context to interpret our results.

To compare our approach with previous works that use multi-input methods requiring multiplexed datasets as training inputs would be of limited utility, since our ExIF approach is explicitly designed to work with 4-plex data inputs (as demonstrated in the EMT application across Figures 3-5) as this makes ExIF accessible to the many researchers who have not yet established multiplexed experimental labelling methods. Hence, there is no like-for-like comparison to be made that is authentic to the intended applications of these different approaches.

Thus, to reiterate, we have altered this section to emphasise: a) the improvement brought by multichannel ‘anchoring’ inputs (now measured by both PCC and SSIM metrics as suggested) as specifically enabled by the ExIF framework, and then; b) to show that different DL architectures are compatible with this approach, thus future-proofing the core ExIF strategy to access future DL model improvements. We later provide comparison to real experimentally multiplexed data as a ground-truth comparison for the performance of ExIF (detailed in **RR1:P5**).

5. [lin. 122-125] It is not surprising that models with more input information (channels) perform better, as seen in 4-plex IF studies. This result should not be presented as specific to ExIF. The manuscript should compare ExIF’s advantages quantitatively with other methods for obtaining multichannel (multi-marker) data.

RR1:P5

The extensible labelling design in Figure 1A is a core part of ExIF. It allows models to be trained with multiple input channels, whilst remaining within the constraints of 4-plex imaging, which is readily accessible to most fluorescence imaging practitioners. This design improves both virtual image fidelity and downstream analysis performance

(classification and manifold learning), highlighting the extensible labelling design as a key component of the ExIF framework. Though multichannel inputs are possible by other means (and may also improve performance), to date such approaches first rely on experimentally multiplexed training data (as discussed in **RR1:P0**), meaning that these approaches are less accessible than 4-plex dependant ExIF and not comparable on a like-for-like basis.

Whilst reiterating our point from **RR1:P4** that, due to the differences in training data dependencies (4-plex inputs for ExIF vs experimentally multiplexed inputs for previous multiplexed generation methods), comparison with previous multiplexed virtual generation methods is not interpretable or 'like-for-like', we completely agree that a comparison to the true performance of experimentally multiplexed raw data (i.e. the 'ground-truth' method for obtaining multichannel data) is valuable.

Accordingly, we have now included a comparison to real experimentally multiplexed data in Supplementary Figure 2 (generated specifically for this revision to compare ExIF with cyclic immunofluorescence (cyclicIF) of 13 markers plus DIC in A549 cells spanning control, EGF and TGF- β conditions). This experiment mirrors as closely as possible the design of the main 4-plex 'EMT' dataset shown in Figures 3-5 – within the constraints of using a slightly different antibody marker panel due to availability of directly conjugated primary antibodies for cyclicIF. In addition to basic fidelity comparisons (Supplementary Figure 2 A&B), this figure compares classification performance for real multiplexed data (Supplementary Figure 2C, bottom row) *versus* various configurations of ExIF-derived multiplexed data (which mirror the configurations presented for the 4-plex integrated dataset in the main text and Figures 3-5).

This new comparison shows, as expected, that while full ExIF achieves far greater performance than standard 4-plex labelling (whilst only being dependent on standard 4-plex data inputs), true experimental multiplexing achieves the highest possible classification accuracy. Similarly, comparison of manifold learning performance to compare cell phenotype heterogeneity (Supplementary Figure 2D) and pseudotemporal marker dynamics inference (Supplementary Figure 2E) show that ExIF again achieves massive improvements over standard plexity datasets, approaching but not quite matching true experimentally multiplexed data.

These results confirm that improvements due to additional information/channels are not specific to ExIF, but can also be achieved (as expected) through real experimental multiplexing – where that capability has been established by researchers. Nonetheless, we believe that this new comparative data supports the strong utility of ExIF for users who need to perform multiplexed analyses within the constraints of low (e.g. 4) plexity imaging, while confirming that true experimental multiplexing is still the optimal approach, where it is possible.

We note that, after careful consideration, we have placed the above comparison as a supplementary figure for two reasons (rather than, for example, simply replacing the original data and presentations in Figures 3-5). First, the model training process is complex to communicate, since although all cells in the dataset are actually labelled with all markers, it is necessary to ignore markers in an arbitrary pattern in order to train models in a manner that emulates the 4-plex ExIF concept. Whilst we have done this, we are concerned that, if this were the only form of presentation for an application of ExIF, this may needlessly confuse some readers.

Second, and extending on this concern, because this dataset and the dataset used in Figure 2 are both actually experimentally multiplexed datasets used herein as sources of complete ground-truth comparison, we are concerned that readers may conclude that ExIF actually requires experimentally multiplexed inputs for training, which is absolutely not the case. To ensure that this misinterpretation is avoided, we have retained our presentation of the original ExIF example that is actually based on 4-plex labelling in the EMT context (Figures 3-5), since this is a direct implementation of the ExIF concept reliant only on 4-plex input data, i.e. as ExIF is intended to be used.

[Results]

6. [lines 136-137]: The claim that ResViT is the first transformer-based architecture for virtual labeling should be reviewed considering recent studies (e.g., DOI: 10.1109/TMI.2020.2968504 and DOI: 10.1117/1.JBO.29.3.036004).

RR1:P6

We appreciate the correction and have removed this claim. This claim is not central to our messaging, since the ExIF strategy is confirmed to be compatible with various DL model architectures, so users can select architectures suited to their needs as these methods evolve in future.

7. [lines 137-143]: Figure 2A-left-B-C-D-E should be presented separately as it pertains to the development and testing of computational elements but does not constitute a functional phase of ExIF itself. Combining these elements in one figure makes it harder to follow the explanation. Additionally, it would help to clarify the connection between this figure and the process shown in Figure 1.

RR1:P7

We have amended the figure to focus only on the relative performance of label-free vs multichannel inputs, using only the ResViT DL model. As discussed in **RR1:P0** and **RR1:P4**, comparison of DL model performance is no longer discussed, with U-Net and

cGAN models shown only in the supplement, and only to support the compatibility of the ExIF framework with various DL model architectures for virtual labelling. As suggested by the reviewers, we have made this clear in the text, adding (line 154) – *“We therefore begin by comparing virtual labelling quality between label-free image-based predictions and those augmented with one or more fluorescence channels as inputs (schematised in Figure 2A), testing computational elements of ExIF.”*

8. [lin. 157-159]: While experiments using only the DIC channel as input show an advantage for ResViT, this does not guarantee that the differences between architectures will remain significant when more input channels are used. With more input data, all architectures should perform better, potentially reducing the observed differences and even their statistical significance.

RR1:P8

We appreciate the feedback. As detailed in **RR1:P4**, we have removed direct comparisons between architectures. We have also added new data analyses showing performance improvements with both U-Net and cGAN architectures when including multichannel inputs (Supplementary Figure 1).

9. [lines 166-172]: In the virtual labeling experiment, input from a DIC channel and seven markers is used to predict an eighth marker, but it is not clear which marker is inferred. Clarify this.

RR1:P9

We present the aggregate results from all possible combinations of inputs to outputs. We have clarified this in text (line 179): *“Extending this multichannel input strategy to the limit of our 8-plex dataset, we built ResViT (Figure 2C) as well as U-Net and cGAN models (Supplementary Figure 1A) incorporating DIC plus seven fluorescence marker channels as input to predict the eighth marker not included as input (in all possible combinations).”*

10. [lin. 236-238] Clarify whether ExIF generates a single model for virtual labeling of all EM state markers or multiple models, one for each marker. Indicate whether pre-trained models were fine-tuned with data from each experiment. This explanation could be detailed in the methods section.

RR1:P10

As noted in our overview of the DL model training approach (updated as per **RR1:P2**) in Figure 1C and on line 89, we give the following general description indicating that a single DL model is trained for each variable channel: *“While this initially gives rise to non-*

integrated data comprising multiple 4-plex IF image sets spanning various channels and cell populations (Figure 1B), the anchoring channels are then used as consistent inputs for multiple generative DL models that are each trained to produce ‘virtual labelling’^{11,12} of a single variable channel (Figure 1C). Since anchoring channels are (by design) present in all cell populations, all virtual labelling models (one per variable channel) can then be applied in all cell populations, thus integrating every variable channel marker into every cell.”

To address the specific context of this reviewer comment, we have also added clarification in the context of model training in the EMT datasets, as follows (line 214):
“We then performed data integration via virtual labelling, using the recurring general markers and DIC as inputs to ResViT virtual labelling models to predict labelling for variable (non-recurring) EM state markers, following the virtual labelling and dataset integration schema depicted in Figure 1C&D. Note that one DL model is trained for each (variable) EM marker. Training was performed using images from three wells per variable marker (one well per treatment condition) to finetune models that were pretrained using ImageNet21K.”

11. [lin. 262-264] It should be clarified to what extent the superior performance obtained is attributed to the use of the ExIF dataset, rather than being an emergent property of the capacity of generative virtual labelling models (regardless of the method followed to obtain the dataset).

RR1:P11

We confess to being slightly unclear as to the interpretation of this question. We assume that the reviewer is referring to the input dataset as the ExIF dataset and whether or not differences in the input dataset significantly alters performance or whether generative virtual labelling models perform similarly regardless of the input data.

To address this question, we highlight multiple lines of evidence indicating that input data significantly affects performance. First, we have compared different image inputs into virtual labelling models, some of which do not take full advantage of the optimal extensible labelling design in ‘Full ExIF’ – that is DIC + three anchoring channels as input. For instance, when we compare virtual image fidelity (Figure 2) or downstream task performance (Figures 4 and 5) we see significant improvements for full ExIF compared to using DIC images only as DL model inputs. Second, in Figure 2D, we show explicitly that addition of different individual input markers to predict specific targets results in variable performance – reaffirming that input data matters to virtual labelling performance. Thus, the performance improvements observed with Full ExIF are both a product of the Full ExIF input dataset and the generative virtual labelling. In other words, generative

virtual labelling alone does not achieve equivalent performance when used irrespective of its input images.

12. [lin. 520-522] Review the statements made so that they are justified with arguments based on quantitative comparisons with the alternatives mentioned in the text (i.e., related works).

RR1:P12

Thank you for this suggestion. We have adjusted our claims (line 458), as follows:

“Taken together, we find that ExIF integration – even when using only label-free (DIC) inputs – can achieve analytical outcomes that outperform interrogation of standard IF alone or the 3 general marker data, despite in some comparisons being based on that same exact underlying image data. Moreover, our full ExIF approach, which achieves high-fidelity integration to create high-plexity fluorescence imaging data, enables powerful multimolecular analyses of discrete cell states, continuous phenotypic heterogeneity and (pseudotemporal) marker dynamics. Achievable with just a single experiment comprising any desired number of standard 4-plex IF labelling conditions (and hence any arbitrary number of integrateable markers), these results approach, though do not yet match, the performance of actual experimentally multiplexed data - whilst maintaining accessibility to the large number of fluorescence imaging practitioners that routinely use standard 4-plex fluorescence imaging techniques.”

Based on our experiments, we have shown that ExIF provides greater biological resolution than that afforded by direct analysis of experimental 4-plex data only, as exemplified in the downstream analytical tasks shown in Figures 4 and 5. Additionally, we have made comparisons between the ExIF approach using multiple fluorescence plus label-free image inputs *versus* standard label replacement using label-free image inputs only (performance compared in the same datasets throughout this manuscript: Figure 2C; Figure 3C&D, Supplementary Figure 1A&B, Supplementary Figure 2B). It is thus shown that ExIF significantly outperforms the standard label replacement approach. While our study does not draw non like-for-like comparisons to previous multiplexed generation strategies due to their dependencies on real multiplexed training inputs, as noted in **RR1:P4**, we have compared extensively to the prevailing approach for label replacement that relies on label-free inputs.

In terms of direct comparison to alternative methods for multiplexed data production, we again note that we have added comparison to experimentally multiplexed data as detailed in **RR1:P5**. This confirms, as expected, that ExIF approaches but does not match experimentally multiplexed data, and we have made this performance comparison clear throughout the paper as detailed in **RR2:P0**.

[Discussion]

13. [lin. 585-586] Include a concise list of the innovations claimed by the authors and explain why they are considered novel in the context of virtual labeling.

RR1:P13

We have added the following summary (lines 532) – *“In contrast to previous virtual labelling, ExIF is end-to-end, relying only on a single experiment/dataset for training and prediction, integrating markers present into a multiplexed dataset beyond the plexity restrictions of the original data. This makes ExIF unique, not limited in plexity¹¹⁻²² or requiring initial experimental multiplexing for model training²⁴⁻²⁶.”*

[Methods]

14. [lin. 634-645] Ensure that detailed information is provided regarding the datasets used in the experiments with A549 and DU145 cells. It would be helpful to present this information in a tabular format.

RR1:P14

We appreciate the feedback. We have added a summary of the details of the dataset in Supplementary Table 3.

15. [lin. 672-673] It is especially important to highlight the computational cost required for the application of ExIF since this value allows a fair comparison with other alternative methods of obtaining multimodal datasets. That is:

- The total time spent in a study carried out (described in the sections contained in line 227-522) should be included following the phases proposed in the workflow using ExIF and this total time should be broken down into its phases. In the training phase of the models, also show the total time spent ¿20 hrs. = 4 hrs./fold x 5 fold?
- Indications on the dependence of the total processing time on the total number of channels (markers) used in ExIF.
- Differentiate between the times required when using label-free ExIF and full ExIF application models.
- It might even be highly informative to include the total cost of the hardware equipment used on which the models are trained.

RR1:P15

We have added training time and hardware used to illustrate the computational cost and the cost of the hardware on line 642. *“Model training was conducted using an NVIDIA V100 GPU taking roughly 4 hours per fold for label-free ExIF and roughly 5 hours for full-ExIF resulting in about 20-25 hours for all 5 folds for our dataset. As each additional variable marker requires an additional model, sequential training would scale linearly with the number of variable markers.”*

=====

Secondary issues

1. Review all figures to improve readability (e.g., size), particularly for boxplots, which may benefit from being replaced with tables showing central tendency, dispersion, and t-test values.

We have consulted with the editor, who suggested that we retain graphical rather than tabular presentation of the datasets; a preference of the journal with which we have complied. We have improved readability in all figures wherever possible.

2. Figure titles are often overly detailed and redundant. The titles should not repeat information already explained in the text or within the figure itself.

We have amended figure captions to be more succinct.

3. [Fig. 1-D] The term “Model Training” is used; should “inference” also be mentioned? We have added this.

4. [line 151] The phrase “...Person's correlation...” should be corrected to “...Person's correlation coefficient...”.

We have made the correction.

5. [Fig. 4-C] Recheck the confusion matrices for rounding errors, particularly in the lower rows for Full ExIF (center) and Difference (right).

We have corrected this.

6. [lin. 647] Replace “...virtual training...” with “...virtual labeling...”.

We have made this correction.

7. [lin. 690] The phrase “...no variance assumptions...” should be corrected to “...unequal variance assumption...”.

We have made the correction.

Reviewer #1 (Remarks on code availability):

The code hosted in the GitHub repository is incomplete and only covers simple image

preprocessing and an example of virtual labeling with pre-trained ResViT models. The code does not include the analyses presented in the manuscript. As a conclusion, it is not possible to reproduce the results of the study based on the code provided. However, the confidentially provided datasets do fit the content described in the manuscript.

We have added our code used to perform the analysis on the Github repository.

=====

Reviewer #2 (Remarks to the Author):

=====

The authors present a new method termed "Extensible Immunofluorescence" (ExIF), which utilizes 4-plex immunofluorescence and DIC images to predict additional markers, thereby generating a highly multiplexed dataset. The standard setup described includes three constant markers and one of eight variable markers. The constant markers are used to train a deep learning network to predict the image of the fourth, variable marker. The authors compare the performance of different network architectures (U-Net, CGAN, and ResViT), identifying ResViT as the most suitable for this task. Through a set of control measurements, they assess the accuracy of this approach in predicting features commonly used to characterize single cells in immunofluorescence experiments (Cellprofiler). Furthermore, they apply this method to create a virtual multiplexed dataset of control cells and cells undergoing epithelial-mesenchymal transition (EMT) with EGF or TGF-B1 stimulation. As a final validation, they test their virtual multiplexed dataset in a classification challenge (predicting the cellular condition) and use manifold extraction (PHATE) to illustrate the trajectory of cell states along the EMT axis.

In summary, the authors present a new interesting experimental and computational approach to the multiplexed fluorescence experiment. Although, I would personally question the strong assumption of the authors that multiplexed imaging is complex and inaccessible, I do believe that there is space for additional approaches. The approach presented in this manuscript is a form of clever 'feature engineering'. When the model predicts the virtual channel based on the anchor channels, it is essentially finding a way to represent the information that was already present in those anchor channels in a new form. This process doesn't add new information; rather, it redistributes or reinterprets the existing information into a form that resembles the virtual channel. This is a form of decomposition and can be very beneficial in the downstream analysis.

My primary criticism concerns the tone of the paper. The manuscript is written from the perspective that this approach can fully replace a multiplexed experiment, aiming to prove

its feasibility. In reality, this method will always offer an intermediate level of information in describing a set of cells in a multidimensional space of n markers. It occupies a middle ground between an experiment with a limited number of measured markers and a fully multiplexed experiment. The value of this approach lies in its ability to estimate most markers for most cells using the ExIF method, albeit this comes at the cost of a higher error rate compared to a real measurement in a standard multiplexed experiment. I strongly encourage the authors to include this notion in the manuscript leaving a potential user to evaluate if this new approach is suitable for their needs.

Response to Reviewer 2: Point 0 (RR2:P0)

We thank the reviewer for their valuable feedback.

We agree that improvements in tone were needed and should reflect the relative performance of ExIF compared to full experimental multiplexing. To this end, as suggested, we generated an entirely new experimentally multiplexed EMT dataset (key results presented in Supplementary Figure 2, emulating key results in Figures 3-5 based on the original 4-plex ExIF EMT dataset). This new comparison recapitulates the improvements shown previously with the ExIF approach relative to standard IF, as well as the fact that, as expected, true experimental multiplexing achieves the highest possible performance in downstream biological analyses (classification, manifold learning etc). Rationales for the design and presentation (in the Supplement) of this new multiplexed data are detailed in the response to Reviewer 1 (**RR1:P5**), but are included here also for your convenience:

We note that, after careful consideration, we have placed the above comparison (between ExIF and real multiplexing) as a supplementary figure for two reasons (rather than, for example, simply replacing the original data and presentations in Figures 3-5). First, the model training process is complex to communicate, since although all cells in the dataset are actually labelled with all markers, it is necessary to ignore markers in an arbitrary pattern in order to train models in a manner that emulates the 4-plex ExIF concept. Whilst we have done this, we are concerned that, if this were the only form of presentation for an application of ExIF, this may needlessly confuse some readers.

Second, and extending on this concern, because this dataset and the dataset used in Figure 2 are both actually experimentally multiplexed datasets used herein as sources of complete ground-truth comparison, we are concerned that readers may conclude that ExIF actually requires experimentally multiplexed inputs for training, which is absolutely not the case. To ensure that this misinterpretation is avoided, we have retained our presentation of the original ExIF example that is actually based on 4-plex labelling in the EMT context (Figures 3-5), since this is a direct implementation of the ExIF concept reliant only on 4-plex input data, i.e. as ExIF is intended to be used.

Considering the tone and presentation of the comparison between ExIF and multiplexed labelling, we have made the following changes to the text where we explicitly address this:

Line 34 (abstract): *“Introducing data integration concepts from omics to microscopy, ExIF empowers life scientists to use routine 4-plex fluorescence microscopy to quantitatively interrogate complex, multi-molecular single-cell processes in a manner that approaches the performance of multiplexed labelling methods whose uptake remains limited.”*

Line 138: *“Excitingly, in each case, ExIF-integrated datasets far exceed the performance of the standard 4-plex IF datasets from which they are derived, as well as the performance of label replacement strategies based on label-free image inputs, and ultimately approach (though do not yet match) the performance of true experimentally multiplexed labelling of the same experimental conditions.”*

Line 142: *“Overall, ExIF establishes a novel methodological framework for the computational integration of standard 4-plex fluorescence imaging data, thereby maximising the capacity of such data to illuminate complex cellular processes and responses. ExIF now constitutes a ‘middle-way’ between accessible but analytically constrained 4-plex IF imaging, and the heightened challenge and analytical value of multiplexed labelling methods. ExIF thereby democratises capacity for high-plexity, imaging-based interrogations of complex, multimolecular cell biology using standard 4-plex immunofluorescence methods that are among the most commonly used in cellular and biomedical research.”*

Line 228: *“Finally, because ExIF aims to maximise integrated biological insights using standard 4-plex data inputs, it is important to compare ExIF performance (for downstream biological analyses) to true experimental multiplexing, since this remains – where feasible – the gold standard for imaging-based multimolecular analysis.”*

Line 321: *“Considering the experimentally multiplexed EMT dataset, emulation (by sub-setted training) of 4-plex ExIF-integrated data ($F1 \sim 0.89$) in comparison to true experimentally multiplexed data ($F1 \sim 0.96$) showed that ExIF achieves an accuracy approaching but, as expected, not matching true experimental multiplexing data (**Supplementary Figure 2C**). Nonetheless, the various forms of ExIF all greatly outperformed standard 4-plex ($F1 \sim 0.81$) and 3-plex (general markers; $F1 \sim 0.73$) performance for treatment classification.”*

Line 449: *“Considering results from the experimentally multiplexed EMT dataset, emulation (by sub-setted training) of 4-plex ExIF-integrated data achieves a PHATE manifold structure that closely matches that from the real experimentally multiplexed dataset, though the latter has somewhat higher treatment condition-resolution. By contrast, the Standard manifold structure is poorly resolved and gives rise to a*

pseudotime trajectory with limited interpretability (Supplementary Figure 2D and E). Accordingly, the inference of selected EM marker dynamics based on the Standard manifold shows little correspondence with the dynamics inferred from the manifolds and trajectories based on emulated ExIF and real multiplexing, whose fundamental trends are robust and also highly analogous to each other. Once again, strong correspondence is also observed between the inferred real EM marker dynamics and virtual EM marker dynamics within the ExIF manifold.”

Line 464: *“Achievable with just a single experiment comprising any desired number of standard 4-plex IF labelling conditions (and hence any arbitrary number of integrateable markers), these results approach, though do not yet match, the performance of actual experimentally multiplexed data whilst maintaining accessibility to the large number of fluorescence imaging practitioners that routinely use standard 4-plex fluorescence imaging techniques.”*

Line 485: *“We have demonstrated the end-to-end application of ExIF using a strictly 4-plex ExIF EMT dataset as well as an experimentally multiplexed (13-plex) EMT dataset, which allows direct comparison of emulated 4-plex ExIF integration versus ground-truth experimentally multiplexed data. Consistently, ExIF integrates cells and markers with high fidelity to significantly enrich downstream interrogation of both discrete (via classification) and continuous (via manifold learning) cell phenotype heterogeneity, achieving levels of analytical performance far beyond that of the underlying 4-plex input data, and approaching (though not yet matching) that of experimentally multiplexed data. ExIF thus accessibly enables quantitative, single-cell, imaging-based analyses of multi-molecular cellular states and mechanisms using only routine 4-plex imaging data to nonetheless interrogate highly complex cellular biology.”*

Line 537: *“We have demonstrated major improvements compared to standard IF in capacity to perform several downstream quantitative single-cell, multimolecular analyses. Thus, the ExIF framework achieves a step-change in the utility of standard IF as a method for quantitative single-cell analyses of complex, multimolecular biology, which approaches the levels of analytical performance previously only accessible with multiplexed labelling methods. This democratizes the powerful benefits of high-plexity fluorescence imaging for the thousands of laboratories within which standard IF methods are already routine.”*

In the trajectory inference experiment presented in Figure 5, the authors compare manifolds derived from a standard fluorescence experiment (Standard Manifold) with those from Label-free ExIF and Full ExIF approaches. It is unsurprising that manifolds incorporating more information from the measurement phase perform better.

Response to Reviewer 2: Point 1 (RR2:P1)

We agree that incorporating more information from the measurement phase could naturally lead to improved manifolds. However, in a 4-plex setting where only three markers are concurrently labelled across all cells, the 'standard' manifold represents the practical limit if all cells are to be phenotypically mapped without additional manipulations. As markers are often labelled in parallel panels to circumvent channel restrictions, we see this scenario as realistic and not specifically contrived for ExIF.

To clarify this in the manuscript, we have highlighted this in line 384 - *“Using the 4-plex ExIF EMT dataset, we first constructed a ‘Standard’ manifold using CellProfiler-derived quantitative features from the three recurring general markers, **since these are the only real markers that are present in all cells.**”*

Based on the same configuration of markers, ExIF by design is able to leverage all markers in this setting via computational integration. Our comparison then aims to show the difference in what is practically possible with such a 4-plex dataset, rather than how plexity *per se* influences the analysis. We highlight this in text by adding (line 458) – *“Taken together, we find that ExIF integration – even when using only label-free (DIC) inputs – can achieve analytical outcomes that outperform interrogation of standard IF alone or the 3 general marker data, **despite in some comparisons being based on that same exact underlying image data.**”*

However, a more informative comparison would be between the Full ExIF Manifold and a manifold obtained from a fully multiplexed experiment. Although the authors emphasize the potential to study cellular heterogeneity with the Full ExIF Manifold, it ultimately reveals only a single trajectory, indicating limited complexity. It is plausible that the same conclusion could be reached using three wells (one per condition) in a fully multiplexed experiment.

To elaborate, the experiment currently measures DAPI, two anchor markers, and eight variable markers. In a fully multiplexed experiment, this could be replicated with four rounds of immunofluorescence, using fewer antibodies for anchor markers and the same number of antibodies for variable markers, while requiring fewer cells and less reagent. This would result in a cheaper experimental setup. However, ExIF has the advantage of being completed in a single round, reducing the experimental time by a factor of four. Comparing these two approaches directly is more appropriate than comparing ExIF to a 4-plex fluorescence experiment, and the advantages and disadvantages of the new method should be clearly outlined.

Response to Reviewer 2: Point 2 (RR2:P2)

As detailed more extensively in **RR2:P0**, we have generated a new multiplexed labelling experiment (13-plex; Supp Fig 2) that mimics the experimental conditions (control vs EGF

vs TGF-b) and closely emulates the EM-state labelling panel of the 4-plex EMT dataset presented in Figures 3-5. Using this dataset, we have added a direct comparison of emulated ExIF data (trained using subsetting to mimic 4-plex input data within this dataset) to the same fully multiplexed data, using classification and manifold learning / pseudotime analyses, as per the true 4-plex ExIF analyses presented in Figure 4-5. This new analysis indeed confirms that experimentally multiplexed labelling outperforms ExIF, but that ExIF performance approaches that of full multiplexing when compared to that achievable if analysing raw 4-plex IF input data. This highlights ExIF as a 'middle-way' (line 144) between 4-plex IF and experimentally multiplexed IF, with the advantages of accessibility based on requiring only 4-plex inputs – including for the DL model training phase (unlike other multiplexed virtual labelling approaches). We believe this makes clear that ExIF is not intended as an improvement over experimentally multiplexed measurement, with our comparison to experimentally multiplexed data contextualising the improvements that we report relative to non-multiplexed (4-plex) datasets.

It could be argued that ExIF's ability to analyze a significantly larger number of cells may reveal more complex alternative trajectories or distinct subpopulations. However, it seems unlikely that if such alternative states were not detectable in a single well, the networks generating the missing signals would accurately predict the missing markers to the extent that they would be distinguishable in the final manifold. I would welcome it if the authors could provide evidence to refute this notion.

Response to Reviewer 2: Point 3 (RR2:P3)

We agree that ExIF is not designed to recreate missing signals / populations not adequately contained in the training data. We believe that we don't make this claim, but simply note that ExIF is able to integrate the number of markers and the number of cells acquired across the extensible labelling design into an integrated datasets.

In the clustering analysis presented in Figure 4, the authors use the ExIF dataset to extract cell features, applying PCA and SVM to cluster the data and evaluate the accuracy of cell state prediction. They claim that this experiment "reveals a remarkable increase in classification accuracy due to virtual labelling-enabled data integration, despite both analyses ultimately relying on identical underlying image data (the three general markers)."

However, this statement is not entirely accurate. First, the ExIF analysis includes additional measurements, as the real measurements of the fourth channels were not excluded, based on my understanding. Second, the experiment is designed such that cells within a single image share the same experimental condition. This setup could allow the network to detect correlations in signal levels between cells within the same image,

meaning the virtual channel could encode this information. In contrast, cell classification is performed on a feature set that lacks any information about the cell neighborhood. Consequently, the classification results are confounded by the introduction of cell neighborhood information into the analysis. Therefore, I do not believe this experiment effectively demonstrates the merits of the ExIF approach.

Response to Reviewer 2: Point 4 (RR2:P4)

Thank you for highlighting this, it is an excellent point. We have repeated the analytical experiments including morphology and neighbourhood features as input into the SVM classifier, allowing contextual neighbourhood information to be used in non-ExIF conditions. This represents a typical analysis using the limit of neighbourhood information provided from commonly used pipelines (such as CellProfiler). As such, our comparison demonstrates the practical limits of typical single cell feature based analyses using standard 4-plex data vs ExIF integrated data, but we simultaneously do our best to minimize any bias imposed by this difference. We have included the following text to highlight this important consideration (line 307): *“Notably, we included CellProfiler-measured spatial and cell morphological features (e.g. cell-cell contact levels and intercellular distances; cell size, shape) capturing local cellular context in all models (Supplementary Table 1), to avoid an imbalanced performance comparison resulting from the potential encoding of spatial information by ExIF models, which would (otherwise) not be available to the SVM classification models containing real image data inputs-only. Thus, the enhanced classification performance of the ‘fluorescence ExIF’ data compared to the ‘general markers’ data appears to be due to the integration of the 8 additional virtual EM markers, mediated by ExIF.”*